# Therapeutic Intervention for Various Hospital Setting Strains of Biofilm Forming *Candida auris* with Multiple Drug Resistance Mutations Using Nanomaterial Ag-Silicalite-1 Zeolite

**DOI:** 10.3390/pharmaceutics14102251

**Published:** 2022-10-21

**Authors:** Hanan A. Aldossary, Suriya Rehman, B. Rabindran Jermy, Reem AlJindan, Afra Aldayel, Sayed AbdulAzeez, Sultan Akhtar, Firdos Alam Khan, J. Francis Borgio, Ebtesam Abdullah Al-Suhaimi

**Affiliations:** 1Master Program of Biotechnology, Institute for Research and Medical Consultations (IRMC), Imam Abdulrahman Bin Faisal University, Dammam 31441, Saudi Arabia; 2Department of Epidemic Diseases Research, Institute for Research and Medical Consultations (IRMC), Imam Abdulrahman Bin Faisal University, Dammam 31441, Saudi Arabia; 3Department of Nano-Medicine Research, Institute for Research and Medical Consultations (IRMC), Imam Abdulrahman Bin Faisal University, Dammam 31441, Saudi Arabia; 4Department of Microbiology, College of Medicine, Imam Abdulrahman Bin Faisal University, Dammam 40017, Saudi Arabia; 5Department of Pathology & Lab Medicine, King Fahad Specialist Hospital, Dammam 32253, Saudi Arabia; 6Department of Genetic Research, Institute for Research and Medical Consultations (IRMC), Imam Abdulrahman Bin Faisal University, Dammam 31441, Saudi Arabia; 7Department of Biophysics Research, Institute for Research and Medical Consultations (IRMC), Imam Abdulrahman Bin Faisal University, Dammam 31441, Saudi Arabia; 8Department of Stem Cell Research, Institute for Research and Medical Consultations (IRMC), Imam Abdulrahman Bin Faisal University, Dammam 31441, Saudi Arabia; 9Institute for Research and Medical Consultations (IRMC), Imam Abdulrahman Bin Faisal University, Dammam 31441, Saudi Arabia

**Keywords:** *Candida auris*, biofilm, hydrothermal, nanomaterial, Ag-silicalite-1, therapeutics, multidrug-resistant mutations, drug-resistant genes

## Abstract

*Candida auris* (*C. auris*), an emerging multidrug-resistant microorganism, with limited therapeutical options, is one of the leading causes of nosocomial infections. The current study includes 19 *C. auris* strains collected from King Fahd Hospital of the University and King Fahad Specialist Hospital in Dammam, identified by *18S rRNA* gene and *ITS* region sequencing. Drug-resistance-associated mutations in *ERG11, TAC1B and FUR1* genes were screened to gain insight into the pattern of drug resistance. Molecular identification was successfully achieved using *18S rRNA* gene and *ITS* region and 5 drug-resistance-associated missense variants identified in the *ERG11* (F132Y and K143R) and *TAC1B* (H608Y, P611S and A640V) genes of *C. auris* strains, grouped into 3 clades. The prophylactic and therapeutic application of hydrothermally synthesized Ag-silicalite-1 (Si/Ag ratio 25) nanomaterial was tested against the 3 clades of clinical *C. auris* strains. 4wt%Ag/TiZSM-5 prepared using conventional impregnation technique was used for comparative study, and nano formulations were characterized using different techniques. The antibiofilm activity of nanomaterials was tested by cell kill assay, scanning electron microscopy (SEM) and light microscopy. Across all the clades of *C. auris* strains, 4 wt%Ag/TiZSM-5 and Ag-silicalite-1 demonstrated a significant (*p* = 1.1102 × 10^−16^) inhibitory effect on the biofilm’s survival rate: the lowest inhibition value was (10%) with Ag-silicalite-1 at 24 and 48 h incubation. A profound change in morphogenesis in addition to the reduction in the number of *C.*
*auris* cells was shown by SEM and light microscopy. The presence of a high surface area and the uniform dispersion of nanosized Ag species displays enhanced anti-*Candida* activity, and therefore it has great potential against the emerging multidrug-resistant *C. auris*.

## 1. Introduction

Disease is an abnormal situation, physically and pathophysiologically. Disease has been classified into infectious or non-infectious diseases. Infectious communicable disorders group is the main pathogens leading to death globally. Antibiotic treatment is the major therapy against bacterial infection, but its abundant and unrealizable targets and the deficiency of new antibiotics and vaccines are main causes of the increase in resistance of infectious bacteria [1]. This practice on antibiotics use helps greatly in single drug-, multidrug-, and total drug-resistant contagious bacteria, which proliferate as baleful and/or deadly strains in the host, either Gram-positive or -negative bacteria that attach any part of the body’s digestive, excretory, and respiratory systems as well as causing blood sepsis, cystic fibrosis, skin contagious, and teeth inflammation [2,3,4,5,6]. Bacteria improve their resistance against antibiotics by different mechanisms: one of them is reducing the permeability of the cell membrane, the inactivation of enzyme, target protection or overproduction, changing the receptor site, and rise outflow as a result of the over-expression of efflux pumps [1]. The efflux pump is a biological pump mechanism that drives the antimicrobial drug to the outside of the microbe, in addition to the inactivation of porin channels that suppress the drug’s entry to the microbial cells. As a result, bacteria change to multidrug-resistant bacteria (MDR) [7,8,9]. Biofilm matrices is another complex mechanism which is the phenotype of the three-dimensional accumulative gathering of microbes since cells usually cement to polymeric substances in the network of extracellular space that mostly contains polysaccharides and certain proteins, in addition to exterior nucleic acids. The biofilm is a permanent phenotype that supports the microbe with the capability to counter drugs and antibiotics [10,11]. The formation of biofilm by multidrug-resistant microbes is correlated to the high tolerance of antibiotics by bacteria [1]. The β-lactamase enzyme acts essentially to damage the β-lactam ring in a β-lactam group of antibiotics. So, the efflux pump, porins, biofilm and β-lactamase are the main phenotypes involved in developing MDR in microbes [8,9,11,12]. Once the antibiotic resistance suppresses the antibiotic inhibitory effect on pathogenic bacteria, these resistant bacteria proliferate with the treatment by antibiotics [13,14]. 

MDR pathogens are serious and outstanding risks globally, which creates an urgent need for novel bioeffective substitutions for fighting aggressive MDR attacks [15]. Thus, the antibiotic resistance issue is one of the major public health problems [1] since the health of humans is dramatically influenced by the critical rise in the resistance modalities of antimicrobials against detrimental bacteria. Based on the declaration by the World Health Organization (WHO) and CDC (Centers for Disease Control and Prevention) that the world is moving to a post-antibiotic era, the prediction of mortality caused by contagious bacteria has risen in comparison to cancer [1,16]. Additionally, according to WHO, the world load of healthcare-related infections varies amidst (7%–12%). So, the screening of device-associated nosocomial infections (DANIs) by healthcare providers is an important issue. Shahbaz et al. [17] used nanotechnology strategy to estimate the prevalence of microbial contaminations in DANIs patients and evaluated the in vitro prevention of MDR bacterial strains. The increase in antibiotic resistance and the deficiency of novel antimicrobial drug attracts many initiatives to develop higher effective antimicrobial strategies for developing new drugs, delivery systems, and targeting management. 

Several strategies are developed to cope with MDR. Nanoparticles (1–100 nm) have been applied as an antimicrobial novel drug to act as antimicrobial effective agents or as delivery systems to bacterial infection sites. Nanoparticles, such as carbon nanotubes, inorganic and organic, may abolish the drug-resistance phenotype in bacteria. The nanotechnology is associated with its antimicrobial activity and ability to suppress biofilm accumulation [1]. In recent years, a nanocomposite as a biocompatible oil-in-water cross-linked polymer has been developed, in which the nanocomposite degrades in some physiological environment, demonstrating its ability to enter, eliminate wide-spectrum of MDR bacteria, and expel biofilms without toxic side effects in the in vitro study of a mammalian fibroblast cell line. It was noticed that sequential passaging prevented bacteria from developing resistance to the used nanocomposites, showing promising potential for a degradable nanocomposite effect on MDR microbes [18]. Another strategy is the combined use of nanoparticles with plant-based antimicrobials to beat the possible toxic effect and to suppress the resistance modalities by bacteria, including efflux pumps, formation of biofilms, mediation to quorum sensing; and for probably plasmid treatment [1]. Although the therapeutic nanotechnology against MDR introduces promising results, it still presents challenges. So, many studies have been conducted to investigate nanomaterials as a new therapeutic system to fight MDR microbes. Ref. [11] synthesized nanomaterials, studied the way of drug resistance in superbug *P. aeruginosa*, and evaluated the nanocomposites as an anti-pseudomonal factor and to act as a drug-resistant reversal tool, in addition to the mode of action of these composites, as well as the nanomaterials’s druggability. Another antimicrobial nanotechnology was to use the fungal culture of (*Mucor circinelloides*), extract the chitosan, and convert it to nanochitosan to synthesize selenium bioactive nanoparticles (SeNPs) immediately, utilizing the extract of *Hibiscus sabdariffa* (Hb), and to evaluate their biocidal actions against MDR bacterial pathogens. The chitosan of fungi had 86.71% deacetylation and converted to nanochitosan with a size of 67.6 nm. The SeNPs were biosynthesized directly by dexterity interaction with Hb; the average size of Hb/SeNPs was 12.1 nm. The coupling between nanochitosan and Hb/SeNPs was successful. The fabricated nanocomposites showed increased antibacterial effect against all studied MDR infectious microbes; the nanochitosan/Hb/SeNPs composite was very robust with the largest growth inhibiting zones with minimum doses of bactericidal. The structure of nanochitosan with Hb-biologically synthesized SeNPs resulted in the potent concept of integrated bactericidal effect toward MDR microbes with biosafety, eco-care, and efficiency. A very recent study was designed to figure out the antibacterial effect of a nanosheet complex compound, and zinc oxide nanoparticles, solely and also mixed with specific antibiotics, against *Pseudomonas aeruginosa* isolate. These nanocompounds suppressed the MIC of tetracycline from 16 to 64 times against the MDR clinical isolate. The mode of action of the nanosheet was two synergistic effects, including interference with efflux pumps, and blocked biofilm synthesis. Furthermore, these nanosheet and nanocomposites decreased the mutant prevention concentration of TET [11]. 

The emerging application of nanoparticles can be applied against mortal bacterial infections. So, different notions of nanomaterial technology can overcome the challenge of antibiotic resistance. The use of nanoparticles assists in the invention of fabricated antimicrobial nanotherapeutics by affirming of the functionality and delivery of a nanoparticle’s exterior design and fabrication for antimicrobial load [16]. One recent study was on 324 patients diagnosed with DANI. Biosynthesized nanocomposites were analyzed for their antimicrobial activity. A total of 369 bacterial pathogens was isolated from DANI patients. A ratio of 87% of these microbes was Gram-negative bacilli and all were MDR. A total of 41.5% of the Gram-negative isolates were ESBL maker. Among the Gram-positive bacteria, methicillin-resistant Staphylococcus aureus represented a ratio of 7.3% of the overall isolates. The nanocomposites exhibited a dose-dependent effect, since 100% had a bactericidal effect at a concentration of 5 mg/mL during 3 h in incubation media, while a decreased treatment with 2.5 mg/mL took 6 h to suppress perfect growth. Nanocomposites are an alternate treatment to block the DANIs on *Acinetobacter baumannii* and *Citrobacter*, the most causative species [17].

The genus *Candida* is one of the most known types of infectious yeast. Among them, *C. auris* is an emerging multi-drug resistance (MDR) pathogen with high mortality rates in humans [19,20,21]. It belongs to the *Clavispora* clade of the *Metschnikowiaceae* family that commonly infects human blood circulation, digestive system, skin, wounds, and other organs. *C. auris* is mostly resistant to antifungal agents, resulting in a mortality rate of up to 60% [19]. *C. auris* has been reported in over 20 countries worldwide [22], and is the most emergent MDR fungal pathogen with >500 incidences of infection from the Arabian Peninsula [23]. The property of *C. auris* to form biofilms makes it critical to deal with and treat [24]. Moreover, during this ongoing COVID-19 pandemic, clinical cases of *C. auris* as a secondary infection have been reported. Long hospitalization for critical cases of COVID-19 and treatment with antimicrobial-drug may be a cause of clinical infections of *C. auris* [25]. Phenotypically, inaccurate diagnoses of *C. auris* as Candida famata, Candida haemulonii, and Candida catenulata delay contagion prevention and dissemination. *C. auris* diagnosis is difficult by using microbiology identification techniques [26,27,28,29]. Among techniques used to reveal *C. auris*, the molecular method could lead to the accurate identification of *C. auris* to the species class. Kordalewska and collaborator [30] prepared a real-time PCR assay that helps in *C. auris* detection and differentiation from other *Candida* spp., such as *C. haemulonii*, *C. lusitaniae* and *C. duobushaemulonii*.

Whole-genome sequencing (WGS) confirmed the isolates that were observed in six continents and classified them into four clades based on the geographic region of *C. auris* isolates (South Africa, East Asia, South Asia, and South America clades). Recently, the fifth potential clade was described in the Iran region [31]. WGS of *C. auris* shares about 78%–85% similar location with *C. glabrata* and *C. albicans* genes. *FKS1, FKS2, FKS3 ERG3* and *ERG11* genes were observed for the presence of various mutations in *C. auris* [32,33]. Antifungal agents play a main role in inhibiting the mechanism of fungal cell wall synthesis or fungal cell membrane synthesis, as fungal cells are wrapped in carbohydrates, which are essential for the growth and fungi survival, for instance, β (1,3)-D-glucan synthases machinery (GS) and chitin synthases machinery (CS). Despite the drug agent effect as a treatment, the efficiency of an antifungal is weak due to the drug’s low solubility, which needs to enhance the bioavailability in the target tissues by frequent dosages. This resulted in an increase in drug–drug interactions, as well as the possibility of antifungal resistance and toxic side effects in host tissues [34,35].

The sensitivity of *C. auris* to azole drugs is related to the point mutations in *lanosterol 14 α-demethylase (ERG11)* gene. Ergosterol is an essential component in the synthesis of the fungal cell membrane by using lanosterol 14-alphademethylase. Antifungal drugs significantly inhibit the *ERG11* enzyme, consequently, inhibiting the cell membrane pathway [36,37,38] as well as resulting in the gene overregulation of efflux pump encoding genes [39,40]. In addition, antifungal resistance to amphotericin B has been linked to different mutated *ERG* genes [29].

Lockhart et al. [41] observed mutations in *C. auris* and specific genetic clades frequently described among fluconazole-resistant isolates, specifically in the *ERG11* gene, with F126L, Y132F, and K143R mutations. Rybak et al. [42] demonstrated a novel genetic mutation in the transcription factor, the zinc-cluster transcription factor-encoding gene *(TAC1B),* which was associated with fluconazole resistance and could be produced speedily after the isolates exposed the fluconazole [38]. Rhodes et al. [43] reported that the *F211I* amino acid substitution in the *FUR1* gene was associated with resistance to flucytosine (5-fluorocytosine). 

Antifungal therapy used in healthcare is limited and is mostly based on azole agents (triazoles and imidazoles), lipopeptides (echinocandins) and polyenes (amphotericin B, mycostatin (nystatin) and natamycin). However, drug resistance, low bioavailability (5% absorption) and the toxic effects of drug are the limitations in therapies [44,45]. One of the potential treatments against *C. auris* infection could be Ag-based nanotherapeutics. Kamli et al. [46] found that the nanocomposite of (Ag-Cu-Co) has the capability to be an antifungal drug against infectious *C. auris*. Recently, Vazquez-Munoz et al. [47] reported that silver nanoparticles (Ag NPs) are active against both the planktonic cells and biofilm. Zeolites are microporous aluminosilicates consisting of Si, Al and O species in the framework along with exchangeable cations (Na, Mg, K, Ca, etc.), that determine the antimicrobial activity [48].

Silicone rubber/Ag/Zeolite has been found to show a long-term fungistatic effect. An increase in the zeolite content enhanced the antimicrobial effect [49]. Ag ion-exchange zeolite in composite with polyurethane and silicone rubber are found to exhibit antimicrobial activity. The polymer and Ag/zeolite ratio tends to influence the antimicrobial activity, polymeric physio-chemical and thermal properties [50]. Ag nanoparticle dispersion, the availability of free Ag species, and functional moieties are critical in the antimicrobial effect [51]. However, several reported antimicrobial nanomaterials, their synthesis techniques and expected microbicidal activity are far from the clinical treatment demands. The efficacy, biocompatibility and ease of multifunctionality are desired. In addition, the developing cost of nanomaterials also needs to be considered for clinical-scale applications. Silicalite-1 is the siliceous form of ZSM-5, which has been widely used as an adsorbent in industrially relevant processes [52]. Such a non-acidic form of zeolite is not much explored in medical applications. Particularly, Ag-incorporated high silica zeolite synthesized through in situ hydrothermal conditions has not been explored as a microbicide agent. 

In the present study, we have developed Ag-silicalite-1 nano-formulation as an effective antibiofilm agent against the emerging drug-resistant *C. auris*, as the biofilm is one of the modes of action used by the microbe to resist the antibiotics. So, Ag-silicalite-1 was synthesized, characterized with different physicochemical techniques involving phase determination, surface texture, template effect and morphological characterizations using XRD, BET, TGA-DTA, SEM and TEM. The technique is simple, reproducible, and scalable. 

With the increased prevalence of microbial contaminations in DANIs patients, caused by many microbes, *C. auris* is the most important multidrug-resistant fungus. Then the anti-biofilm activities of the designed Ag-silicalite-1 nano-formulation can be used as a surface sterilizer against *C. auris* in the future. As it is known, *C. auris* contaminates many surfaces, devices and tools in the centers of healthcare providers, which increases its infectious effect among the already or newly admitted patients and worsens their status. Only a few works have studied *C. auris* as the most virulent infectious microbe. Therefore, in this study, the authors isolated it from different hospitals and treated it with the novel nano- formulation of Ag-silicalite-1 as an antifungal agent against the isolated 10 *C. auris* strains. 

The synthesized nanomaterials, 4 wt%Ag/TiZSM-5 and Ag-silicalite-1 were tested for their antibiofilm activity. All 19 *C. auris* strains showed a significant inhibition on biofilm survival rate, with a lowest of 10% with Ag-silicalite-1 at 24 and 48 h incubation. Profound morphogenesis and reduction in the number of *C. auris* cells were found by SEM and light microscopy. The presence of high surface area and the uniform dispersion of nanosized Ag species displays an enhanced antifungal activity, therefore having great potential against the emerging drug-resistant *C. auris*. It is clearly revealed that 4 wt%Ag/TiZSM-5 and Ag-silicalite-1 have anti-fungal time- and dose-dependent effects.

## 2. Methodology

### 2.1. Collection and Cultivation of C. auris Strains

Nineteen strains of *C. auris* were collected from microbiology labs from the clinical samples of King Fahd Hospital of the University, Khobar, Saudi Arabia and King Fahad Specialist Hospital, Dammam, Saudi Arabia. The clinical isolates were identified by MALDI-TOF MS analysis at the time of collection. Strains were restored from −80 °C freezer storage and sub-cultured on Sabouraud dextrose broth (SDB) to confirm viability. The broth was incubated at 37 °C for 48 h, and single colonies were isolated using the streak plate method on SDA to confirm the purity. The inoculated agar plates were incubated at 37 °C for 48 h. The slides were examined under the light microscope to describe the morphology of cells.

### 2.2. Molecular Identification

#### 2.2.1. DNA Extraction

Genomic DNA was extracted from 19 strains of *C. auris* using Qiagen’s Yeast/Bact kit (Gentra Puregene Yeast/Bact. Kit, Qiagen’s, Hilden, Germany). The DNA concentration was measured for estimating the DNA quantity and purity using nanodrop.

#### 2.2.2. Amplified of 18S rRNA Gene

A total of 19 strains were amplified by using *18S rRNA* gene (F:5′-GTCTGCAAGTCGTAACAAGGTTTCACTGTAG-3′; R:5′-AAGGAAAGGTCCAGCCGGACCAG-3′) primers. (MoleQule-On, Auckland, New Zealand) at annealing of 61.8 °C in thermocycler (T- professional, Biometra, Gottingen, Germany). PCR amplicons were gel loaded and documented upon electrophoresis. The PCR amplified products were purified using PCR QIAquick purification kit (Qiagen, Germany) for removing the components other than the amplicon. Cycle-sequenced products were sequenced by using 3500 capillary sequencings. Then the amplicons sequenced were aligned in nBLAST. 

#### 2.2.3. Amplified of ITSa and ITSb Regions

A total of 19 strains were PCR amplified and sequenced for the *ITS* regions (ITSaF: 5′-ATTTTGCATACACACTGATTTG-3′; ITSaR: 5′-CGTGCAAGCTGTAATTTTGTGA-3′; and ITSbF: 5′-AGGAATTCCTAGTAAGCGCAAGT- 3′; ITSbR: 5′-ATTTACCACCCACTTAGAGCT-3′) primers. Primers were synthesized at MoleQule-On (Auckland, New Zealand). The amplification of ITSa and ITSb regions were carried out at annealing temperature at 55 °C and 57.5 °C, respectively, in T-professional thermocycler (Biometra, Gottingen, Germany) PCR amplicons, gel loaded and documented upon electrophoresis. The PCR amplicons were purified as described earlier for removing the components other than the amplicon. Cycle-sequenced products were sequenced by using 3500 capillary sequencings. Then the amplicons were sequenced and aligned in nBLAST. All the *ITS* sequences were analyzed using MEGA11 with the reference sequences obtained from NCBI. 

#### 2.2.4. Molecular Analysis Resistance-Associated Mutations

Primers were designed by using primer BLAST at the national center for bioinformatic information tool (NCBI). Resistance-associated mutations in genes, *ERG11* (F primer 5′-ATGGCCTTGAAGGACTGCATCGT-3′; R primer 5′-TTAGTAAACACAAGTCTCTCTTTTCTCCCA-3′)*, TAC1B* (F primer 5′-ACGTGGAGATGAGTCACAGAACGG 3′; R primer 5′ CTTCGCTATCATCAGAATAATTGAGGCAGTT 3′) and *FUR1* (F primer 5′ TGATCCACGAGCTTTAGCGCATCACCTTATC 3′; R primer 5′ AGATGTGGGTCACTCTGAAAGAATATGCTGAAAAC 3′) of 19 strains were amplified by using the PCR technique and visualized using an agarose gel. The PCR amplicons were purified as stated before for removing the components other than the amplicon. Amplicons were sequenced separately by using 3500 capillary sequencing for detecting drug-resistance-associated mutations. A representative of the sequences was submitted to GenBank.

### 2.3. Synthesis of Nanomaterial

In the first step, Ludox (40 wt%) was added dropwise in an alkaline solution and allowed to stir for 15 min. Then, silver nitrate corresponding to Si/Ag ratio 25 (0.33 g) dissolved in 2.5 mL water was added and stirred for 15 min. Then tetrapropyl ammonium hydroxide (TPAOH (40%)) was added dropwise and stirred for 1.5 h. Then the solution mixture was hydrothermally treated for 3 days. After the treatment, the precipitate was filtered, dried, and calcined to obtain Ag-silicalite-1. 

For TiZSM-5, silica source Ludox AS-40 and template TPAOH (40%) were mixed and stirred for 1 h. Then, 0.63 mL of TIP (titanium isopropoxide) was added and stirred for 10 min. Then, we added the required amount of aluminum nitrate, dissolved in 2.5 mL distilled water, and stirred for 15 min. The above mixture was then added to an alkaline solution (2.5 M NaOH) and stirred overnight. Then the solution was aged for 48 h and then hydrothermally treated, filtered, dried and calcined. 

Then, 4 wtAg was impregnated over TiZSM-5 using the wet impregnation technique. A total of 0.0126 g of silver nitrate salt was taken and dissolved in 60 mL of water; after dissolution, 2 g of TiZSM-5 was added and stirred overnight. Then the mixture was dried at 100 °C for 24 h and then calcined.

### 2.4. Characterization Analysis of Nanomaterial

The phase of Ag-silicalite-1 was analyzed using benchtop XRD (Miniflex 600, Rigaku, Tokyo, Japan). The textural features, including the BET surface area, pore volume and pore size, were measured using ASAP-2020 plus (Micromeritics, Norcross, GA, USA) based on the nitrogen adsorption technique. The silver nanoparticle chemical coordination was analyzed using DRS-UV-visible spectroscopy analysis (JASCO, Tokyo, Japan). The decomposition of the template of Ag silicalite-1 was analyzed using TGA-DTA (STA 6000, Perkin Elmer, Waltham, MA, USA). SEM analysis of Ag-based ZSM-5 and Ag-silicalite-1 was measured using scanning electron microscopy (SEM) equipped with energy dispersive X-rays (EDX) detector (SEM, FEI, Inspect S50 with 20 kV) and transmission electron microscopy (TEM) (TEM, FEI, Morgagni 268 with 80 kV as working voltage). The silver content was determined using an inductively coupled plasma optical emission spectrometer (ICP-OES) Horiba ULTIMA 2 instrument.

### 2.5. Biofilm Analysis

#### 2.5.1. Cell Kill Analysis

*C. auris* strains, the clinical isolates, were used for the study of nanomaterial affecting biofilm formation for 24 and 48 h. All analyses were performed with replicates, and mean values were considered. To start with, culture cells from preserved stock at −80 °C were revived by streaking onto Sabauraud dextrose agar (SDA) plates and incubating overnight at 37 °C. From the revived plates, a loopful of the strains were inoculated into SDB and further incubated using an orbital shaker at 130 rpm and 37 °C for 24 and 48 h. The Biofilm assay was carried out using the 96-well microtiter plate. Precisely, the cells harvested from overnight SDB cultures were adjusted to a cell density of 10^7^ CFU/mL and diluted. Biofilms were formed on flat-bottom 96-well microplates on incubation at 37 °C for 24 h [53,54]. 

For the 24 h biofilm experiment, the nanomaterials were prepared in a two-fold dilution series in trypticase soy broth (TSB) supplemented with 1% sucrose, and then the dilution series was added to the plates with the *C. auris*, for final nanomaterial concentration ranging from 0.5 to 2 mg/mL for drug Ti-ZSM-5, Ag-silicalite, 4 wt%Ag/Ti ZSM-5 and 0.0250 to 0.0025 mg/mL for AgNO_3_. The untreated culture was considered the growth control. Plates were incubated at 37 °C for 48 h. Post 24 h incubation, the wells were washed two times using phosphate-buffered saline (PBS) to get rid of non-adherent cells. To obtain the firmly attached biofilms, the 96-well plates with PBS were sonicated for 15 min, and the cells were harvested and serially diluted. The selected dilution was evenly spread on the freshly prepared SDB and further incubated for 24 h. Post incubation, plates were manually counted and recorded for the number of cells. Later, the quantity of biofilm formation was determined using a colony-forming unit technique (CFU) and determined as per the following formula: 

A/B×100 (A is the total number of Colonies after treatment, B number of cells in control).

For the 48 h biofilm experiment, the assay was carried out using 96-well microtiter plates. The cells harvested from overnight SDB cultures were adjusted to a cell density of 10^7^ CFU/mL. Biofilms were formed on flat-bottom 96-well microplates by growing *C*. *auris* strains in TSB and incubated at 37 °C for 24 h, then the wells were decanted and added with fresh TSB with the nanomaterial suspended with a concentration ranging from 0.5 to 2 mg/mL for drug Ti-Zsm-5, Ag Zeolite, Ag/Ti Zeolit and 0.0250 to 0.0025 mg/mL for AgNO_3_ and further incubated for 24 h. Post incubation period, the wells were washed two times using phosphate-buffered saline (PBS) to get rid of non-adherent cells. To obtain the firmly attached biofilms, the plates treated with PBS were sonicated for 15 min, and the cells were harvested and serially diluted. Later, the quantity of biofilm formation was determined using the CFU technique [55]. The selected dilution was evenly spread on the freshly prepared SDB, and further incubated for 24 h. Post incubation, plates were manually counted and recorded for the number of cells. Then, the quantity of biofilm formation was determined using the CFU technique and determined as per the following formula: 

CFU/mL = (Number of colonies × dilution factor)/volume of plated culture.

One-way ANOVA analysis was subjected to identify the significance between the treatments and clades. 

#### 2.5.2. Morphogenesis Genesis by Scanning Electron Microscopy Analysis

The effect of the synthesized nanomaterial on the morphogenesis of *C. auris* was examined using SEM. The strains were grown and treated as described in the previous section. Post incubation of 24 h, the cells were harvested, washed, fixed, dehydrated and finally coated with gold. The gold-coated cells were analyzed by SEM [56].

### 2.6. Effect on Planktonic C. auris

The antifungal activity of the nanomaterial on planktonic *C. auris* strains was determined by using light microscopy examination. Briefly, the inoculum was adjusted to a final concentration of 10^6^ cells mL^−1^. Then, *C. auris* strains were added in 96-multi-well round-bottom plates having nanomaterials in a two-fold dilution series in TSB, for a final nanomaterial concentration ranging from 0.5 to 2 mg/mL for drug Ti-Zsm-5, Ag Zeolite, Ag/Ti Zeolite and 0.0250 to 0.0025 mg/mL for AgNO_3_. Plates were incubated at 37 °C for 48 h. Post incubation, the loopful of treated cells and control was placed on a glass slide and observed using a light microscope (Nikon H550L, Tokyo, Japan). The mean value of the duplicates was recorded, and micrographs were taken. One-way ANOVA analysis was carried out to identify the significance.

## 3. Results and Discussion

### 3.1. Molecular Identification of Collected Isolates

All the multidrug-resistant isolates of *C. auris* were re-identified and confirmed using *18S rRNA* and *ITS* regions. All the *18S rRNA* gene sequencers were confirmed as having originated from the *18S rRNA* gene region of *C. auris*. A representative of the sequences was submitted to GenBank (GenBank accession ID: OK001860). All the *18S rRNA* gene sequences were analyzed using MEGA11 with the standard sequences obtained from fungal databases and NCBI. The representative of the results on the phylogenic analysis is shown in Figure 1. The phylogenic analysis via branching diagram or a tree of *18S rRNA* gene sequences with the standard sequences clearly indicates the evolutionary relationships among various *Candia* species with the study isolate of *C. auris* based upon similarities in their *18S rRNA* gene sequences. 

### 3.2. Phylogenetics Based on Its Region

The DNA of the 19 stains of the *C. auris* was PCR amplified with (290 bp) primers (MoleQule-On, Auckland, New Zealand) and sequenced for the *ITS* regions (ITSaF:5′-ATTTTGCATACACACTGATTTG-3′; ITSaR:5′-CGTGCAAGCTGTAATTTTGTGA-3′; and ITSbF:5′-AGGAATTCCTAGTAAGCGCAAGT-3′; ITSbR:5′-ATTTACCACCCACTTAGAGCT-3′). The sequences were aligned in nBLAST, and it was confirmed that the *ITS* of the isolated are from the *C. auris*. All the *ITS* sequences were analyzed using MEGA11 with the sequences obtained from NCBI. The results are presented in the form of phylogeny (Figure 2). 

### 3.3. Drug-Resistance-Associated Mutations

#### 3.3.1. Lanosterol 14-Alpha Demethylase (ERG11)

Each *Candida auris* isolate collected during the study was PCR amplified for the *ERG11* gene with (1575 bp) amplicon primers at 59.4 °C and 62.2 °C annealing temperatures for 35 cycles. All *ERG11* gene amplicons were documented by using 2% agarose gel. All amplicons were purified and sequenced using forward and revised primers, separately. All sequences from the forward and reversed primers aligned in nBLAST and ensured that all are from *ERG11* gene of *C. auris*.

Mutation analysis revealed three variations in the sequence of the *ERG11* gene using the reference sequence, GenBank Accession: KY410388. F132Y and K143R were the missense variants observed in all the strains successfully amplified. Furthermore, only two strains were observed with a single silent mutation (Table 1).

#### 3.3.2. Zinc-Cluster Transcription Factor-Encoding Gene (TAC1B)

*Candida auris* isolates from the study were subjected for the PCR amplification of ***TAC1B*** gene with (693 bp) primers (MoleQule-On, Auckland, New Zealand). It was ensured that all sequences are from the *TAC1B* gene. Mutation analysis revealed six mutations from the strains, including three (*TAC1B:c.1602T > C,p.G534G*; *TAC1B:c.1752T > C,p.Y584Y*; and *TAC1B:c.1755T > C,p.F585F*) silent mutations and three (*TAC1B:c.1822C > T,p.H608Y*; *TAC1B:c. 1831C > T,p.P611S*; and *TAC1B:c.1919C > T,p.A640V*) missense variants (Figure 3 and Figure 4). All the three silent mutations observed in all the multidrug-resistant strains were successfully amplified (Table 2). 

#### 3.3.3. Uracil Phosphoribosyltransferase (FUR1)

All 19 *C. auris* isolates were PCR amplified with (823 bp) primers (MoleQule-On, Auckland, New Zealand) (Figure 5) for *FUR1* gene (F:5′-TGATCCACGAGCTTTAGCGCATCACCTTATC-3′; R: 5′-AGATGTGGGTCACTCTGAAAGAATATGCTGAAAAC-3′). All sequences aligned in nBLAST and it was ensured that all are from the *FUR1* gene of *C. auris*. Mutation analysis revealed no variation in the sequence of the *FUR1* gene using reference sequence, GenBank Accession: CP076749 (Figure 5).

Three groups (clades) of *C. auris* were identified based on the mutation data. Two *C. auris* CA1 and CA14 are grouped as clade 1 with *G534G, Y584Y, F585F, H608Y, P611S, A640V, F132Y, K143R* and *K152K* mutations. Most of the isolates [CA2 CA3, CA4, CA5, CA6, CA7, CA8, CA10, CA11, CA12, CA13, CA15, CA16, CA17, CA18 and CA19] are in clade 2 with *G534G, Y584Y, F585F, H608Y, P611S, A640V, F132Y,* and *K143R* mutations. The remaining strain, CA9 (clade 3) was observed with *G534G, Y584Y, F585F, H608Y, P611S, F132Y,* and *K143R* mutations.

#### 3.3.4. Synthesis and Characterization of Synthesized Nanomaterial

Figure 6A(a–d) shows the X-ray diffraction patterns of TiZSM-5, 4 wt%Ag/TiZSM-5 and Ag-silicalite-1. The analysis revealed a mordenite inverted framework (MFI) structure of ZSM-5 but with a certain level of decrease in crystallinity with metal modification. Ag-silicalite-1 showed the highest crystallinity (71%) and exhibited the presence of cubic shape of Ag NPs. Textural analysis using the nitrogen adsorption–desorption technique is shown in Figure 6B(a–d)). TiZSM-5, 4 wt%Ag/TiZSM-5 and Ag-silicalite-1 samples exhibited a type III isotherm (H3 hysteresis) at an extended relative pressure of 0.8–1.0 (Figure 6B). Ag-silicalite-1 showed the highest BET surface area of 338 m^2^/g with a micro surface area of 215 m^2^/g (Table 3). Pore volume also contained the meso (0.11 cm^3^/g) and micropore volume (0.10 cm^3^/g) with pore size distribution centered at about 2.51 nm. In the case of 4 wt%Ag/TiZSM-5, the impregnation of Ag reduced about 20% of the surface textures of TiZSM-5. 

Coordination of Ag on TiZSM-5 and Ag-silicalite-1 were analyzed using diffuse reflectance spectroscopy (Figure 6C(a–d)). TiZSM-5 showed the presence of two bands at about 215 nm and 260 nm due to the tetrahedral and octahedrally coordinated titanium species, respectively. The extending of the peak width to about 390 nm also indicates the presence of some bulk anatase species. The impregnation of Ag showed no significant changes in the absorbance bands of TiZSM-5. In the case of Ag-silicalite-1, the intensity of the band increases at about 415 nm, indicating the presence of Ag nanoparticles. Such a band is primarily attributed to the presence of free electrons giving rise to surface plasmon resonance.

The TGA-DTA technique was used to study the thermal decomposition of the templated (as-synthesized) form of zeolite to determine the porous superficial composition (Figure 6D). The TGA profile of Ag-silicalite-1 shows a gradual decomposition due to water desorption (up to 150 °C), template TPAOH (up to 350 °C) and silanol condensation to form siloxane (>350 °C) (Figure 6D). The nanomaterial synthesized using the hydrothermal process shows about 20% of the template locked inside the Ag-silicalite-1. The DTA peak shows a broad decomposition peak centered between 150 °C and 700 °C associated with the thermal decomposition of TPAOH interacted with Ag-silicalite-1. The template content shows the facilitation of a large surface area during the synthesis of porous silica. The present synthesis of Ag incorporating into high silica zeolite shows the high surface area of 338 m^2^/g correlating with the high percentage of TPAOH loss observed in the profile of TGA-DTA. 

The morphological features, chemical composition and elemental mapping of two prepared nanomaterials (4 wt%Ag/TiZSM-5 and Ag-Silicalite-1) were analyzed using SEM/EDX (Figure 7a–d). The SEM images show a different morphology of 4 wt%Ag/TiZSM-5 and Ag-silicalite-1 (Figure 7a,c). The 4 wt%Ag/TiZSM-5 shows the presence of nanoclusters in the agglomerated form, while Ag-silicalite-1 shows the regular shaped crystals with average particle size in the range of 600 nm. To investigate the structural features and chemical composition of the prepared nanomaterials, EDX analysis was performed as shown in Figure 7b,d. EDX spectra of Ag/TiZSM-5 revealed the composition of titanium-containing aluminosilicate ZSM-5 zeolite (Ti, O, Si, Al) [57] (Figure 7b). In the spectrum of 4 wt%Ag/TiZSM-5, the presence of the Ti peak indicates the successful incorporation of Ti in the zeolite matrix. The Ag/TiZSM-5 nanocomposite is composed of Ti (3.54 wt%), O (51.78 wt%), Si (32.10 wt%), Al (2.46 wt%), and Ag (0.33 wt%). The appearance of the Ag peak in the Ag/TiZSM-5 specimen is an indication of the possible impregnation of Ag nanoparticles on the zeolite matrix. The EDX spectrum of Ag-silicalite-1 is composed of Ag (4.65%), O (49.87%), Na (1.80%), and Si (33.13%) (Figure 7d). The existence of an Ag peak in Ag-silicalite-1 with a reasonable intensity is an indication of the comprehensive incorporation of Ag nanoparticles within the silicalite-1 matrix.

The elemental mapping examination revealed that all of the constituent elements were equally distributed throughout the specimen powder of the SEM micrographs on the carbon support. It was observed that the main elements were Si and O and Al, which appeared very dense, suggesting the homogeneous preparation of zeolite. The Ti was distributed as well as Ag in 4 wt%Ag/TiZSM, indicating the impregnation of Ag with TiZSM-5. The elemental mapping analysis of Ag-silicalite-1 highlighted the constituent elements of Ag, O, and Si, attributed to the successful incorporation of Ag nanoparticles into the silicalite-1 matrix. The EDX and elemental mapping analyses agree well with each other and support the successful preparation of 4 wt%Ag/TiZSM-5 and Ag-silicalite-1 nanomaterials. 

Further to SEM, TEM was carried out to examine the detailed morphology and structure of 4 wt%Ag/TiZSM-5 and Ag-silicalite-1. The results of TEM are displayed in Figure 8. The morphological analysis shows the intact particles of nanosized Tiand Ag in both samples. The TEM results of 4 wt%Ag/TiZSM-5 revealed the presence of nanoporous, structured, irregular-shaped cluster particles of TiZSM-5 with varied sizes of a few tens of nanometers to a few hundreds of nanometers (Figure 8a,b). The Ag nanoparticles attached to the TiZSM-5 matrix are highlighted with arrows, and the size of the AgNPs was estimated at nearly 10–20 nm. TEM of Ag-silicalite-1 showed regular-shaped large silicalite-1 particles with average size of about 600 nm (Figure 8c,d). The silver nanoparticles intact within the silicalite-1 particles show the successful incorporation of Ag nanoparticles (~10 nm) into the silicalite-1 framework.

### 3.4. Biofilm Studies

The antibiofilm study of synthesized Ti-ZSM-5, 4 wt%Ag/ZSM-5, Ag-silicalite-1 nanomaterials and AgNO_3_ was carried out on 19 *C. auris* hospital strains. The obtained results showed the synthesized nanomaterial displayed potent anticandidal activity by demonstrating antibiofilm activity via CFU killing assay and antifungal effect on the planktonic cells. The biofilm CFU killing assay was performed at two periods of incubation. It was noted that both the studies at different incubations, i.e., 24 and 48 h of incubation yielded a similar activity against the formation of biofilms after treatment with the synthesized nanomaterial. A different rate of biofilm inhibition was seen in each strain compared with their untreated counterpart (control). Irrespective of clades, all strains treated with 4 wt%Ag/TiZSM-5 zeolite and Ag-silicalite-1-nanomaterial showed highly significant inhibition for both biofilm [(24 h; F = 57.4079 and *p* = 1.1102 × 10^−16^) (48 h; F = 43.3833 and *p* = 1.1 × 10^−16^) and planktonic *C. auris,* compared to the control conditions.

Analysis of the effect on mutations on the antibiofilm activity via CFU killing assay and antifungal effect on the planktonic cells revealed no significant impact between clades 1–3. However, AgNO_3_ had a complete inhibitory effect on the biofilm and the planktonic cells of the organism in clade 3, compared to the remaining two clades (clade 1 and clade 2) of the organisms (Figure 9 and Appendix A). Figure 10 shows the representative images of the CFU plates of two *C. auris* strains.

*C. auris* is an emerging pathogenic organism that has gained attention for its widespread resistance to existing antifungal drugs and increasing mortality rates. The formation of biofilm is the basic pattern of most fungi, including *C. auris,* which is significant in developing infections and inhabiting mainly on materials such as implants, including catheters, and hospital settings. Such biofilms are mostly resistant to a number of antimicrobials due to their complex colony-forming ability by producing an extracellular matrix and other regulatory mechanisms, such as quorum-sensing molecules, the upregulation of efflux pump genes, etc. [58,59]. In the current study, the synthesized nanomaterial exhibited significant inhibition on the biofilm production of almost all of the tested *C.auris* strains, which could be due to several factors. The rare combination of the metal ion and the carrier may have a significant impact on their bioaction. In general, the mode of action of the nanomaterial is linked to its particle size and morphology that attaches to specific molecular targets, leading to the disruption of the cell metabolism and cell structure, therefore affecting growth [60]. Ag-doped nanomaterial causes a multitude of simultaneous actions of structural and metabolic disruption in the *Candida* cells, for example, depolarization of the membrane, disruption of cell membrane/wall, elevation in ROS production, and inhibition of enzymatic action, cell arrest, and many more. This collective disruption of cellular organization and function minimizes their ability to resist the nanomaterial action. 

### 3.5. Morphogenesis Study by SEM

The effect of synthesized nanomaterial was also evaluated by studying the morphogenesis of *Candida* cells post-treatment by using SEM. The control cells (Figure 11a) had no effect and appeared normal with smooth cellular surfaces, and treatment with A had also a negligible effect on the cellular integrity (Figure 11b), but (Figure 11c–e) had a significant effect on the morphogenesis of cells. The nanomaterial is clearly seen attached to cell surfaces. This attachment leads to penetration, thereby causing the disruption of the cell wall and cell membrane. The gradual disruption causes the cell to lose cellular integrity and therefore causes cell death.

### 3.6. Effect of Nanocomposite on Planktonic C. auris

The effect on planktonic cells was evaluated by using the light microscope (Nikon H550L, Japan). It was observed that the number of planktonic cells significantly reduced after the treatment, as can be seen in (Figure 12). The pattern of activity for the inhibition of planktonic *C. auris* cells was found similar to that of biofilm inhibition. The nanomaterial Ag-silicalite-1 and 4 wt%Ag/TiZSM-5 showed inhibitory action against the majority of strains, compared to the untreated counterpart. However, TiZSM-5 has no inhibitory effect as also seen in the biofilm assay. Therefore, the activity difference is primarily attributed to the dispersion state and particle size of AgNPs on silicalite-1 and TiZSM-5. To detect the nature of AgNPs species on two supports, we analyzed the XRD pattern of three samples between 2 theta ranging between 30 and 60° (Figure 6A(a–c)). X-ray diffraction analysis showed that Ag nanoparticle dispersion occurs on TiZSM-5 and is less than 10 nm (below detection limit of XRD) over TiZSM-5 (Figure 6A(a,b)). No chunks, such as Ag particles, were observed on the external surface of TiZSM-5, indicating the high dispersion state of Ag on TiZSM-5. This result correlates with the diffuse reflectance spectra of Ag/TiZSM-5 (Figure 6C) due to the reduction in the peak attributed to Ag species with TiZSM-5 support (Figure 6C). SEM-EDX mapping and the TEM profile show the presence of high dispersity (Figure 7 and Figure 8). In the case of Ag-silicalite-1, the XRD pattern confirmed that the AgNPs are crystalline in nature (Figure 6A(c)). Therefore, such crystalline AgNPs on such large crystals of Ag-silicalite-1 is shown to favor the structural and metabolic disruption in *C. auris* cells.

The mechanism of action of Ag-silicalite-1 nanoformulation on drug-resistant *C. auris* as an effective antibiofilm agent may be due to the unique properties of nanomaterials. The unique properties of nanomaterials, which include size, shape, and surface chemistry, generally produce significant inhibitory action on the bacteria. It has been reported that the different sizes and shapes of nanomaterials are analogous to bacterial bio-molecular components, and enhance the better connections that can be controlled through surface functionalization. Moreover, high surface-to-volume ratios of the nanomaterials and multivalent interactions are important aspects for creating antibacterial agents [43,44,45]. Nanomaterials use multiple bactericidal routes and mechanisms, which include direct bacterial cell wall damage, and also the generation of reactive oxygen species (ROS) and binding to intra-cellular bacterial constituents, to successfully kill the bacteria [46]. The mechanisms by which nanomaterials interact with bacteria may be due to the unique physicochemical properties, especially multi-valent interactions with bacterial cells. It has been suggested that Van der Waals forces, receptor-ligand, hydrophobic interactions and electrostatic attractions play a role in nanomaterial–bacteria interfaces for robust antibacterial action [47]. Detailed research studies about the toxicological concerns on the hydrothermally synthesized nanomaterial are needed before using them for applications [61,62]. For future aspects on technological advancements and comparative studies with hyaluronic acid, the use of hydrogels, the role of mitochondria and gap junction proteins’ involvement will be a major area of interest. 

## 4. Conclusions

Present results revealed three groups (clades) of *C. auris* based on the mutation data. Clade 1: Strains with nine (*G534G, Y584Y, F585F, H608Y, P611S, A640V, F132Y, K143R* and *K152K*) mutations. Strains CA1 and CA14. Clade 2: Strains with eight (*G534G, Y584Y, F585F, H608Y, P611S, A640V, F132Y,* and *K143R*) mutations. Strains CA2, CA3, CA4, CA5, CA6, CA7, CA8, CA10, CA11, CA12, CA13, CA15, CA16, CA17, CA18 and CA19. Clade 3: Strains with seven (*G534G, Y584Y, F585F, H608Y, P611S, F132Y,* and *K143R*) mutations. Strain CA9. Additionally, there were two hot-spot amino acid substitutions (encoding *F132Y* and *K143R*) in the *ERG11* gene in all strains as a missense mutation, which were reported as being fluconazole resistant [63]. Mostly, (encoding *K143R*) mutations were highly geographic associated with India and Pakistan clades [41]. These findings are from a study in 2018, which reported that (encoding *F132Y* and *K143R*) mutations might be fluconazole resistant in South Asian and American clades [64]. Moreover, there were only 2 strains out of the 19 observed amino acid substitutions (encoding K152K) as single silent mutations in the *ERG11* gene. For all the sequences, it was ensured that 19 strains were found to harbor six mutations from the *TAC1B* gene. Three (encoding *G534G*, *Y584Y* and *F585F*) were revealed as silent mutations, and three were observed (encoding H608Y, P611S and A640V) as missense variants. Thus, these data show that mutations in the *TAC1B* gene represent all isolates with *TAC1B* mutations had fluconazole resistance and confirm the roles of the *S611P* (*TAC1B*) mutations in the azole resistance of *C. auris* [42,65]. The synthesized nanomaterials, 4 wt%Ag/TiZSM-5 and Ag-silicalite-1 were tested for antibiofilm activity, and all 19 *C. auris* strains showed a significant inhibition of the biofilm survival rate irrespective of the clades, with the lowest being 10% with Ag-silicalite-1 at 24 and 48 h incubation. Insignificant differences among the clades with various mutations on the inhibiting activities of synthesized nanomaterials, 4 wt%Ag/TiZSM-5 and Ag-silicalite-1, indicate the highly consistent effect of synthesized nanomaterials for further studies. However, the complete inhibitory effect AgNO_3_ on the organism in clade 3 shall not be neglected during the next stages of analysis in the future. Profound morphogenesis and reduction in a number of *C. auris* cells were found by SEM and light microscopy. The presence of the high surface area and uniform dispersion of nanosized Ag species display enhanced antifungal activity and therefore have a great potential against the emerging drug-resistant *C. auris*. 

It is widely known that multidrug-resistant microbial contaminations in patients with device-associated nosocomial infections are prevalent and caused by a variety of microorganisms, including *C. auris*, the most common multidrug-resistant fungus. In the future, the anti-biofilm properties of the developed Ag-silicalite-1 nano-formulation might be exploited as a surface sterilizer against *C. auris*. We briefly discussed current advances in employing the aforesaid nanomaterials to attack *C. auris* in antifungal applications in this paper. However, there is still more work to be done before these materials can be used in therapeutic settings. The nanoparticles’ precise mechanism, biological effects, and toxic and side consequences remain unknown. We still do not know if nanoparticles and their metabolites are actually harmless to the environment and biology. To address all of these issues, further laboratory and clinical trials are needed, but nanomaterials’ promise as antifungal therapy is undeniable.

## Figures and Tables

**Figure 1 pharmaceutics-14-02251-f001:**
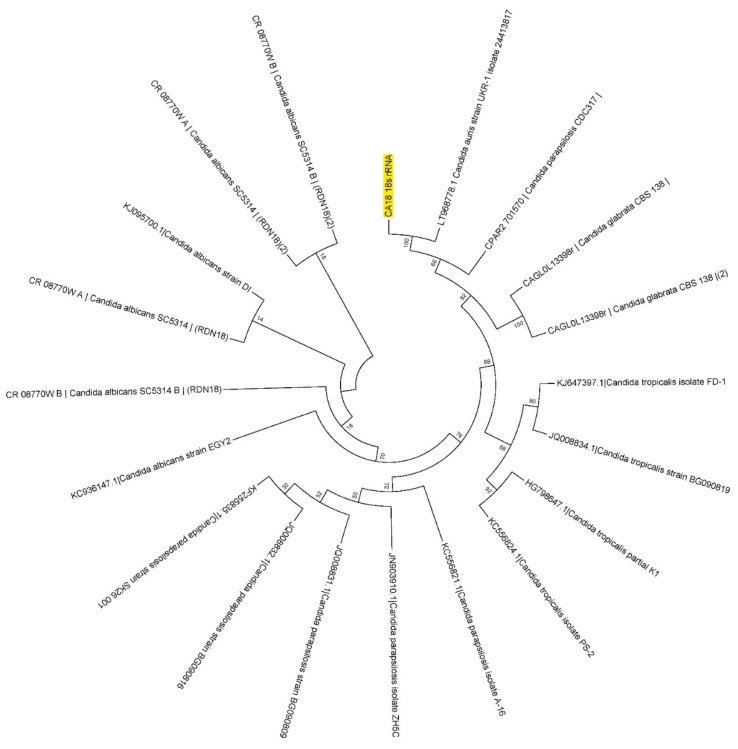
Representative of the phylogenetic analysis of the *Candida auris* isolates. The isolate CA18 from the *Candida auris* isolates analyzed using the *18S rRNA* sequence by maximum likelihood method and phylogenetic tree constructed using MEGA7. The yellow highlight indicates the *C. auris* CA18.

**Figure 2 pharmaceutics-14-02251-f002:**
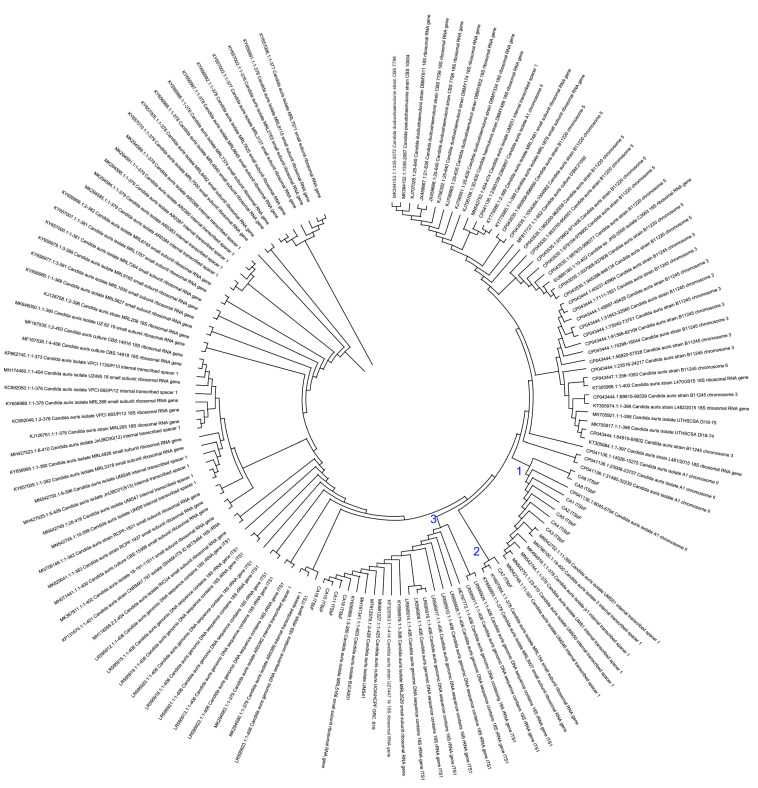
ITS sequenced based phylogenetic analysis of the *Candida auris* isolates. The blue-colored numbers indicate the three clades of the *Candida auris* isolates. Phylogenetics tree using ITS sequence constructed using MEGA11.

**Figure 3 pharmaceutics-14-02251-f003:**
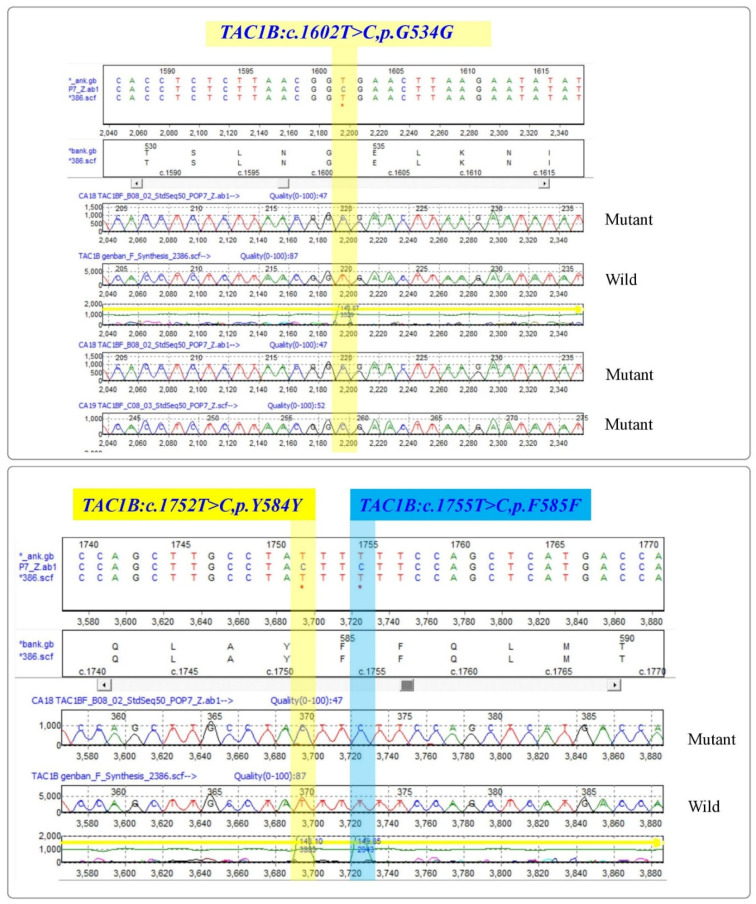
List of silent mutations identified in the zinc-cluster transcription factor-encoding gene (*TAC1B*) of *Candida auris* isolates. Top: *TAC1B:c.1602T > C,p.G534G*; Bottom left: *TAC1B:c.1752T > C,p.Y584Y*; Bottom right: *TAC1B:c.1755T > C,p.F585F*. Reference sequence: GenBank Accession: MW368409.1. * indicates change of nucleotide at this position.

**Figure 4 pharmaceutics-14-02251-f004:**
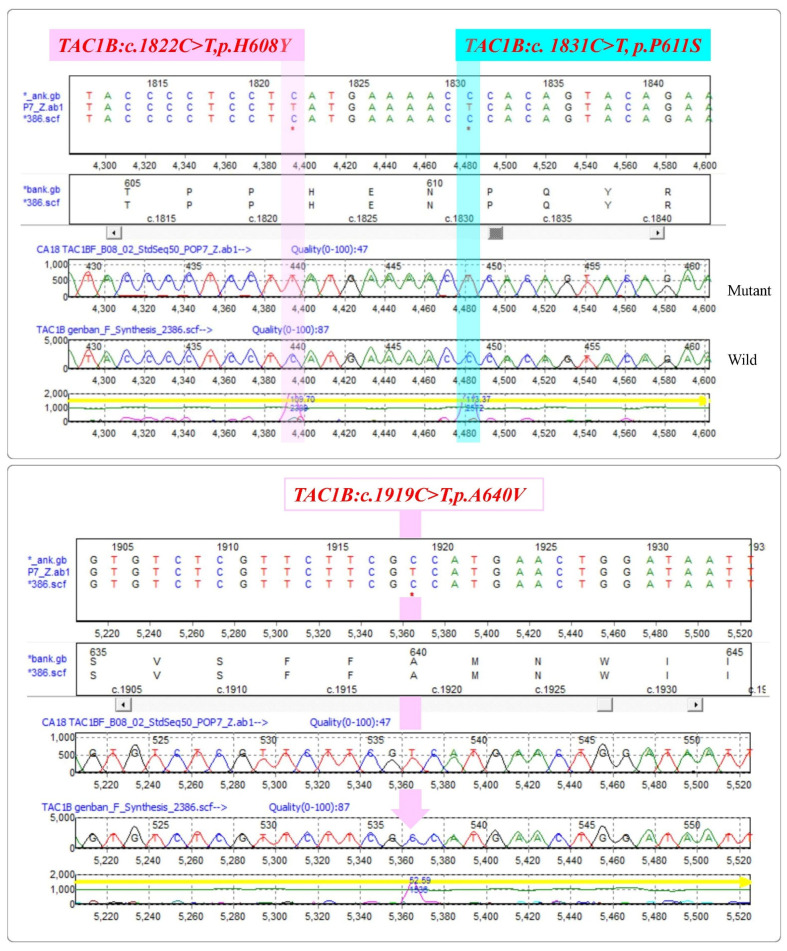
List of missense variants identified in the zinc-cluster transcription factor-encoding gene (*TAC1B*) of *Candida auris* isolates. Top left: *TAC1B:c.1822C > T,p.H608Y*. Top right: *TAC1B:c. 1831C > T,p.P611S*. Bottom: *TAC1B:c.1919C > T,p.A640V*. Reference sequence: GenBank Accession: MW368409.1. * indicates change of nucleotide at this position.

**Figure 5 pharmaceutics-14-02251-f005:**
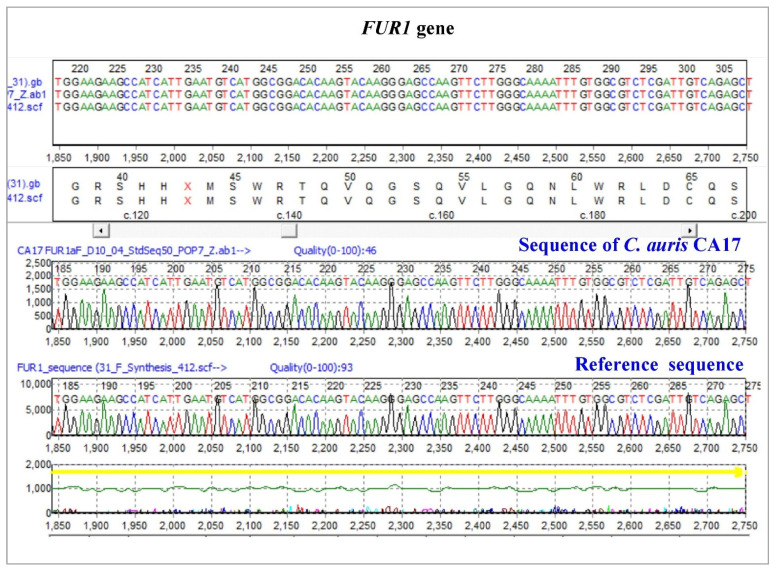
Representative sequence for *FUR1* gene of *C. auris* with no variation.

**Figure 6 pharmaceutics-14-02251-f006:**
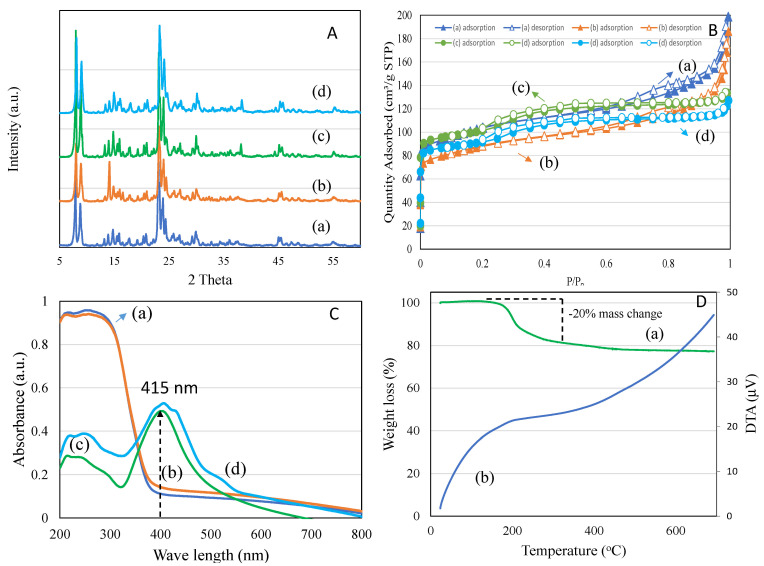
(**A**) X-ray diffraction (**a**) TiZSM-5, (**b**) 4 wtAg/TiZSM-5, (**c**) Ag-silicalite-1 (Ag/Si ratio 25) and (**d**) Ag-silicalite-1 (Ag/Si ratio 100), (**B**) N2 adsorption isotherm of (**a**) TiZSM-5, (**b**) 4 wtAg/TiZSM-5, (**c**) Ag-silicalite-1 (Ag/Si ratio 25) and (**d**) Ag-silicalite-1 (Ag/Si ratio 100), (**C**) diffuse reflectance spectra of (**a**) TiZSM-5, (**b**) 4 wtAg/TiZSM-5, (**c**) Ag-silicalite-1 (Ag/Si ratio 25) and (**d**) Ag-silicalite-1 (Ag/Si ratio 100) and (**D**) TGA-DTA measurement of as-synthesized Ag-silicalite-1.

**Figure 7 pharmaceutics-14-02251-f007:**
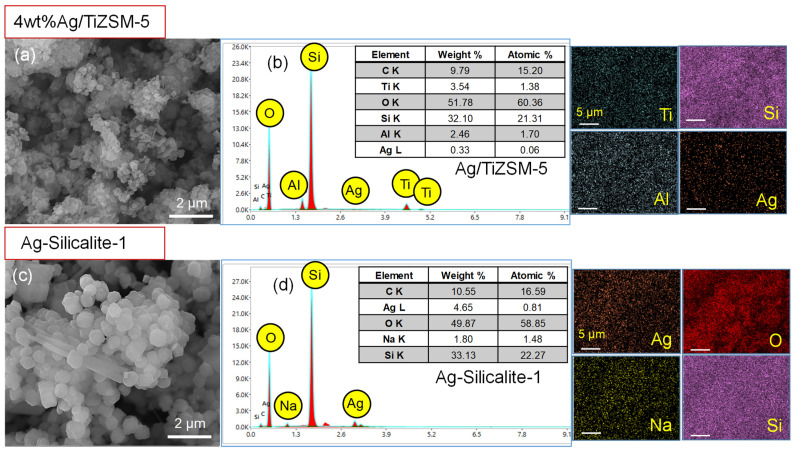
SEM, EDX spectra and EDX mapping images of Ag impregnated TiZSM-5 (**a**,**b**) 4 wt%Ag/TiZSM-5 and (**c**,**d**) Ag-silicalite-1. The scale bars are 5 µm in EDX mapping images.

**Figure 8 pharmaceutics-14-02251-f008:**
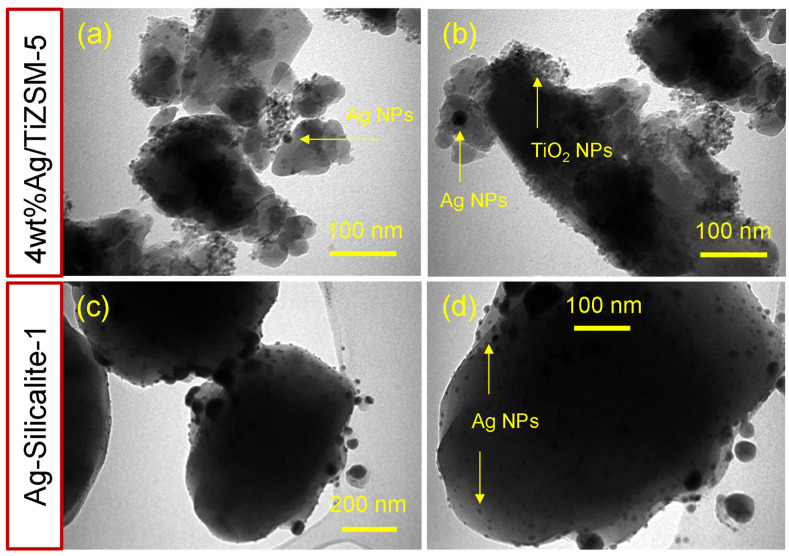
TEM images of Ag impregnated TiZSM-5 (**a**,**b**) 4 wt%Ag/TiZSM-5 and (**c**,**d**) Ag-silicalite-1.

**Figure 9 pharmaceutics-14-02251-f009:**
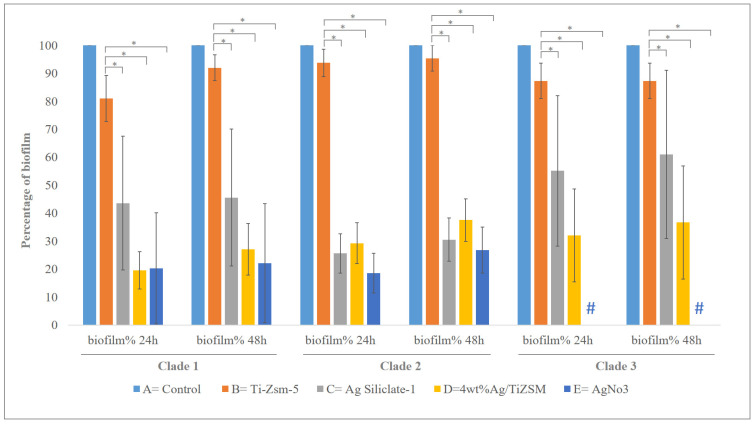
Graph showing the effect of synthesized nanomaterial on the formation of biofilm of 3 clades of *C. auris* strains after 24 and 48 h incubation period. # Complete absence of biofilm. * *p* < 0.01.

**Figure 10 pharmaceutics-14-02251-f010:**
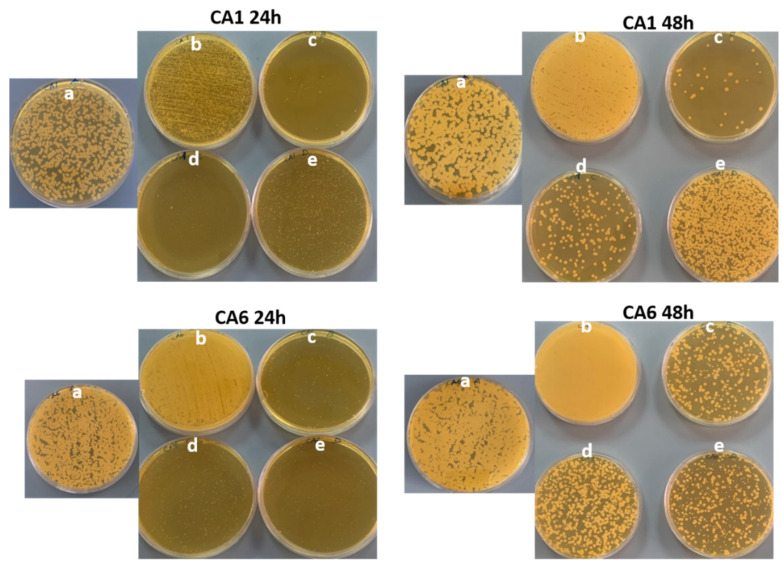
Representative agar plate images of two *C. auris* strains showing the effect of synthesized nanomaterial on the biofilm formation obtained by CFU technique. (**a**) Control (untreated); (**b**) Ti-ZSM-5; (**c**) Ag-silicate-1; (**d**) 4 wt%Ag/TiZSM-5; (**e**) AgNO_3_.

**Figure 11 pharmaceutics-14-02251-f011:**
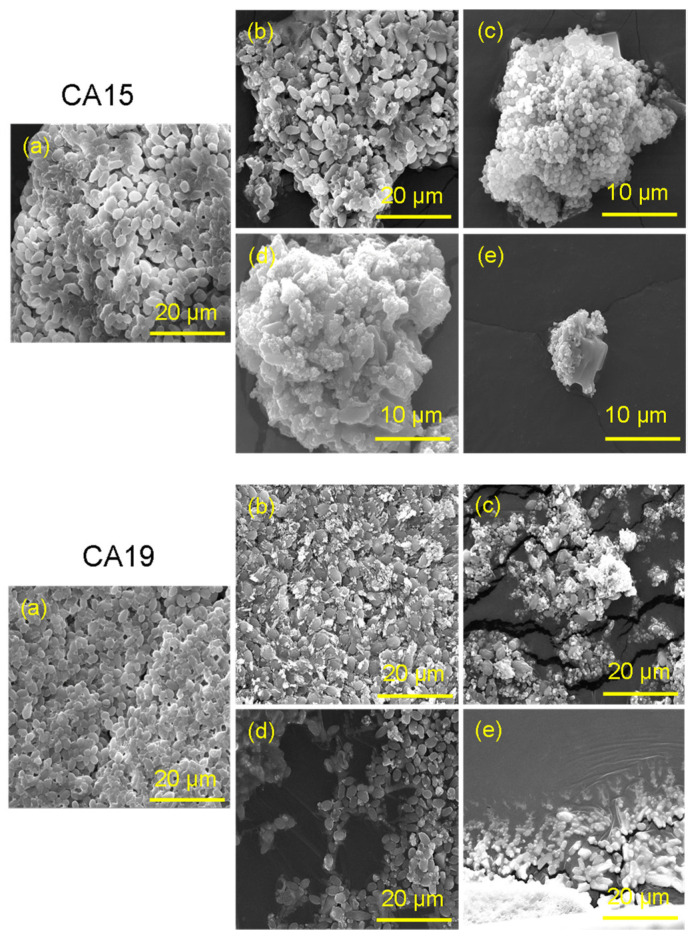
SEM micrographs showing the morphogenesis effect after treatment with nanomaterial. (**a**) Control (untreated); (**b**) Ti-ZSM-5; (**c**) Ag-silicalite-1; (**d**) 4 wt%Ag/TiZSM-5; (**e**) AgNO_3_.

**Figure 12 pharmaceutics-14-02251-f012:**
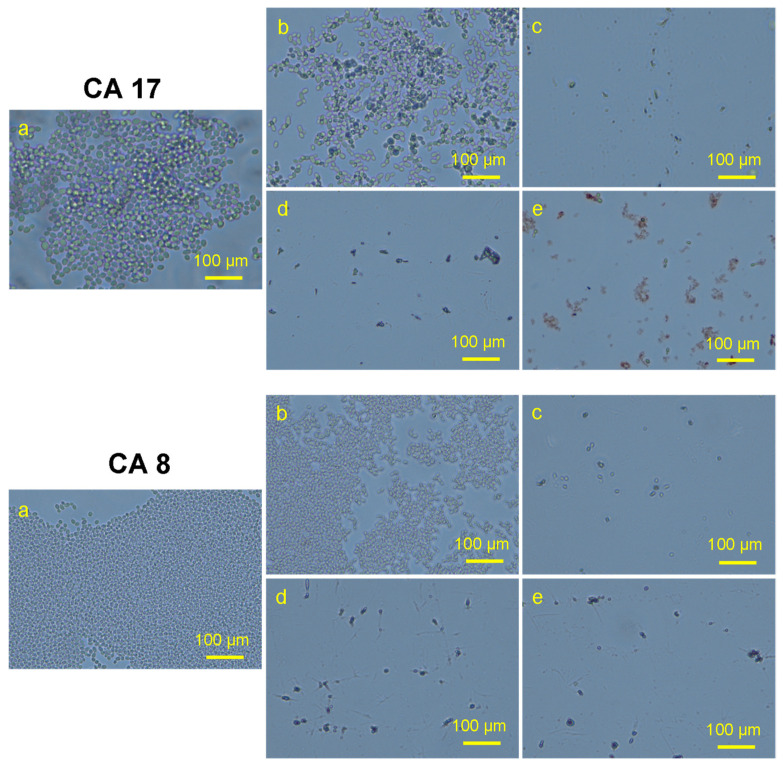
Representative microscopic slide pics of two *C. auris* strains showing the effect of synthesized nanomaterial on the planktonic cells using a light microscope. (**a**) Control (untreated); (**b**) Ti-ZSM-5; (**c**) Ag-silicate-1; (**d**), Ag/Ti zeolite; (**e**) AgNO_3_.

**Table 1 pharmaceutics-14-02251-t001:** List of variants identified in the *lanosterol 14-alpha demethylase (ERG11)* gene of *Candida auris* isolates.

Strain No.	*F132Y*	*K143R*	*K152K*
CA1	+ve	+ve	+ve
CA2	+ve	+ve	-
CA3	+ve	+ve	-
CA4	+ve	+ve	-
CA5	+ve	+ve	-
CA6	+ve	+ve	-
CA7	+ve	+ve	-
CA8	+ve	+ve	-
CA9	+ve	+ve	-
CA10	+ve	+ve	-
CA11	+ve	+ve	-
CA12	+ve	+ve	-
CA13	+ve	+ve	-
CA14	+ve	+ve	+ve
CA15	+ve	+ve	-
CA16	+ve	+ve	-
CA17	+ve	+ve	-
CA18	+ve	+ve	-
CA19	+ve	+ve	-

- negative; +ve positive for mutation.

**Table 2 pharmaceutics-14-02251-t002:** List of missense and silent variants identified in *TAC1B* gene of *Candida auris* isolates.

	Silent Variants			Missense Variants		
Strain No.	*G534G*	*Y584Y*	*F585F*	*H608Y*	*P611S*	*A640V*
CA1	+ve	+ve	+ve	+ve	+ve	+ve
CA2	+ve	+ve	+ve	+ve	+ve	+ve
CA3	+ve	+ve	+ve	+ve	+ve	+ve
CA4	+ve	+ve	+ve	+ve	+ve	+ve
CA5	+ve	+ve	+ve	+ve	+ve	+ve
CA6	+ve	+ve	+ve	+ve	+ve	+ve
CA7	+ve	+ve	+ve	+ve	+ve	+ve
CA8	+ve	+ve	+ve	+ve	+ve	+ve
CA9	+ve	+ve	+ve	+ve	+ve	-ve
CA10	+ve	+ve	+ve	+ve	+ve	+ve
CA11	+ve	+ve	+ve	+ve	+ve	+ve
CA12	+ve	+ve	+ve	+ve	+ve	+ve
CA13	+ve	+ve	+ve	+ve	+ve	+ve
CA14	+ve	+ve	+ve	+ve	+ve	+ve
CA15	+ve	+ve	+ve	+ve	+ve	+ve
CA16	+ve	+ve	+ve	+ve	+ve	+ve
CA17	+ve	+ve	+ve	+ve	+ve	+ve
CA18	+ve	+ve	+ve	+ve	+ve	+ve
CA19	+ve	+ve	+ve	+ve	+ve	+ve

-ve negative; +ve positive for mutation.

**Table 3 pharmaceutics-14-02251-t003:** Textural characteristics of ZSM-5 modified samples.

Sample Code	Surface Area (SA) (m^2^/g)	t-Plot MicroSA (m^2^/g)	MesoSA (m^2^/g)	Pore Volume (cm^3^/g)	t-Plot MV (cm^3^/g)	MesoPV (cm^3^/g)	PSD (nm)
TiZSM-5	329	161	*168*	0.29	0.08	0.21	3.55
4 wt%Ag/TiZSM-5	*267*	131	*136*	0.26	0.07	0.19	3.97
Ag-silicalite-1	338	215	*123*	0.21	0.10	0.11	2.51

## Data Availability

All data are included already.

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
