# Peer review of "Therapeutic Intervention for Various Hospital Setting Strains of Biofilm Forming Candida auris with Multiple Drug Resistance Mutations Using Nanomaterial Ag-Silicalite-1 Zeolite"

_pharmaceutics, 2022, doi:10.3390/pharmaceutics14102251_

Round 1
Reviewer 1 Report (Previous Reviewer 4)
The authors have revised the manuscript carefully. However, some points need more consideration. 1) Fig. 2 was not clear. 2) The English must be improved.
Author Response
Thank you for your effort and time.
As per the comments by the respected reviewer, we have replaced the old figure 2 with new high definition figure 2 in the revised manuscript. We have taken care of the English in the revised manuscript.
Reviewer 2 Report (New Reviewer)
1.It is suggested to add new research studies about the toxicological concerns.
2.what is the suggestion of this study for future works?
3.Please discuss about the using of new biomaterials with new technologies including nanoantioxidants, also please compare your results with hyaluronic acid used hydrogels.
4.It will be better to add the role of mitochondria and gap junction proteins
5.More references for the discussion part of manuscript and bold your study novelty should be added: e.g.,
-DOI: 10.3389/fbioe.2022.855136
-DOI: 10.1155/2022/7211066
Author Response
Thank you for your valuable comments. As per these suggestions, we have added the following content in the discussion to reflect the “new research studies about the toxicological concerns”, “suggestion of this study for future works”, “new biomaterials”, and “role of mitochondria and gap junction proteins” in the revised manuscript. Furthermore the suggested references have also been cited in the revised manuscript as follows.
“Detail research studies about the toxicological concernson the hydrothermally synthesized nanomaterial are needed before using them for applications 64,65. For future aspects on technological advancements and comparative studies with hyaluronic acid used hydrogels and role of mitochondria and gap junction proteins involvement will be of major area of interest.”
64. Baran, A., Fırat Baran, M., Keskin, C., et al., Investigation of antimicrobial and cytotoxic properties and specification of silver nanoparticles (AgNPs) Derived from cicer arietinum L. green leaf extract. Front. Bioeng. Biotechnol. 2022: 855136.
65. Keskin, C., Baran, A., Baran, M.F., et al., Green Synthesis, Characterization of Gold Nanomaterials using Gundelia tournefortii Leaf Extract, and Determination of Their Nanomedicinal (Antibacterial, Antifungal, and Cytotoxic) Potential. J. Nanomater. 2022: 7211066.
This manuscript is a resubmission of an earlier submission. The following is a list of the peer review reports and author responses from that submission.
Round 1
Reviewer 1 Report
It appears that publication in Pharmaceutics would be premature at this time.
This work presents the development of Ag-silicalite-1 nanoformulation as an effective antibiofilm agent against drug-resistant C. auris stains. Although the paper is focused on a very serious health concern and the project is interdisciplinary, the concept is not clearly presented, the novelty is not supported in the whole manuscript, while the results are not well-presented and analyzed in the respective part.
Beginning from the Introduction,
- this section is not well-structured, and the part about the research background in the field of antimicrobial nanomaterials for MDR strains is too small, as opposed to the first part describing the auris clades and techniques used for classification of several strains.
Many questions need to be addressed in the Intro such as:
- What is the significance of this study?
- What practical value does this study have?
- What are the advantages of current work over previous antimicrobial strategies?
- It is mentioned that “The efficacy, biocompatibility and ease of multifunctionality are desired.” However, it is not explained if the proposed nanomaterial has all these characteristics.
- In the last paragraph of Intro, all the above questions need to be answered.
- Moreover, there are too many low-level errors, such as different fonts (first paragraph of Intro), missing spaces (last paragraph of intro) etc. that should not appear in a submission.
Materials & methods
- This section offers all the info needed for reproducing the results. However, there are many typing errors, such as: 24 h, 24h or 24 hrs??? Is it 37°C or 37 °C??? The font is not the same in every part.
Moving on to the Results & Discussion part,
- The “Collection of isolates” is suggested to be incorporated in the Materials and method part.
- the figures 1,2,3,4,5 are not clearly shown. The data presented cannot be interpreted and the manuscript lacks a discussion about those results.
- The “Synthesis and characterization” part is obviously written from another person; thus, the results are presented in different manner in one paper. The authors used many techniques to characterize their samples, but they did not explain their results appropriately. For example, they used EDX mapping analysis and they did not highlighted in that section the uniformly dispersed Ag NPs.
- On the other hand, in Figure 10 the arrows show Ag NPs, which are not uniformly dispersed. The authors need to add an explanation for that.
- In the biofilm studies, the graphs presented in figure 11 need statistical analysis. Otherwise, expressions such as “depicted a significant inhibition” should be removed.
- Why are two time points presented; Since the activity is similar between 24 h and 48 h, there is no reason for presenting both of them (move one of them in SI).
- The morphogenesis study by SEM needs arrows or something in the figures to highlight all the changes written in the text.
In the conclusion part,
- the significance of the study should be clearly written. This part presented in this manuscript looks like an abstract, not a conclusion.
References
- Authors are suggested to include more updated references reflecting the diversity of antimicrobial nanomaterials and strategies against MDR microbes. Hence, readers can learn the research background comprehensively and the latest progress in the related field.
Author Response
Reviewer 1
It appears that publication in Pharmaceutics would be premature at this time.
This work presents the development of Ag-silicalite-1 nanoformulation as an effective antibiofilm agent against drug-resistant C. auris stains. Although the paper is focused on a very serious health concern and the project is interdisciplinary, the concept is not clearly presented, the novelty is not supported in the whole manuscript, while the results are not well-presented and analyzed in the respective part.
Response:
Authors thank the first reviewer for the valuable suggestions and comments. These suggestion are very helpful to improve our manuscript
Beginning from the Introduction,
- this section is not well-structured, and the part about the research background in the field of antimicrobial nanomaterials for MDR strains is too small, as opposed to the first part describing the aurisclades and techniques used for classification of several strains.
Response:
Thank you for your valuable comment
A new section of three pages with additional references has been added to explain the background of the field of antimicrobial nanomaterials for MDR strains.
Many questions need to be addressed in the Intro such as:
- What is the significance of this study?
Response:
Thank you for your important comment
- auris, the most important multidrug resistant fungus in the study, which was proved practically by the presence of drug resistant mutations for its multidrug resistance. Also, in the present study, we have designedAg-silicalite-1 nano-formulation to act as potential and effective nano-formulationto block the biofilm produced by the emerging drug-resistantC. auris.The authors isolated nineteen C. auris strainswhich were found in the hospitals. As the biofilm is one of the modes of action used by the microbe to resist the antibiotics. So, Ag-silicalite-1 was synthesized, characterized with different physicochemical techniques involving phase determination, surface texture, template effect and morphological characterizationsto be applied as a potential anti-biofilm agent against C. auris
- What practical value does this study have?
Response:
Thank you for your valuable comment
It is well known that there is an increased prevalence of microbial contaminations in Device Associated Nosocomial Infections (DANIs) patients that caused by many microbes, in which C. auris,the most important multidrug resistant fungus. Then the anti-biofilm activities of designedAg-silicalite-1 nano-formulation can be used as surface sterilizer against C. aurisin the future. As it is known that C. auriscontaminates many surfaces, devices and tools in centers of health care providers and increase its infectious effect among the already or new admitted patients and worsens their status.
- What are the advantages of current work over previous antimicrobial strategies?
Response:
Thank you for this critical comment
Only few number works studied on C. auris the most virulent infectious microbe. Therfore, this study isolated it from different hospitals and treated it with novel nano- formulationofAg-silicalite-1 as antifungal agent against the isolated nineteen C. auris strains.
- It is mentioned that “The efficacy, biocompatibility and ease of multifunctionality are desired.” However, it is not explained if the proposed nanomaterial has all these characteristics.
Response:
Thank you for your valuable comment
The synthesized nanomaterials, 4wt%Ag/TiZSM-5 and Ag-silicalite-1 were tested for its antibiofilm activity. All the nineteen C. auris strains showed a significant inhibition on biofilm survival rate, with a lowest of 10% with Ag-silicalite-1 at 24 and 48h incubation. Profound morphogenesis and reduction in a number of C. auris cells were found by SEM and light microscopy. The presence of high surface area and uniform dispersion of nanosized Ag species displays an enhanced antifungal activity and therefore having a great potential against the emerging drug-resistant C. auris.
- In the last paragraph of Intro, all the above questions need to be answered.
Response:
Thank you for your comment. We have added all the above answers inthe revised manuscript as follows.
“Disease is anabnormal situation physicallyand pathophysiologically. Diseasehas beenclassified intoinfectious or non-infectiousdiseases. Infectious communicable disordersgroup isthe mainpathogensleading to death globally. Antibiotics’ treatmentis the majortherapyagainst bacterial infection, but itsabundantand unrealizabletargets, deficiencyof newantibiotics and vaccinesare main causes of increasetheresistance ofinfectious bacteria(Baptistaet al, 2018).This practice onantibiotics use helps greatly in single drug-, multidrug-, and total drug-resistant contagious bacteria which proliferate as baleful and/ ordeadlystrain in the host either gram-positive and -negative bacteriathat attach any part of the body digestive, excretory, respiratory systems as well as blood sepsis, cystic fibrosis, skin contagious, teeth inflammation. (Martins et al, 2013, Mohr, 2016, Karaiskoset al, 2019 Khardoriet al, 2020, and Mirzaeiet al, 2020). Bacteria improved its resistance against antibiotic by different mechanism, one of them isreducepermeabilityof cellmembrane, inactivation of enzyme, target protectionor overproduction,changereceptorsite, riseoutflowas a result toover-expression of efflux pumps(Baptistaet al, 2018). The efflux pump is a biological pump mechanism that drives outthe antimicrobial drugtothe outsideof the microbe, in addition to the inactivationof porin channels that suppressthe drug’s entry to the microbial cells. Finally, bacteria changes to multidrugresistant bacteria(MDR) (Xavieret al, 2010and Fernández, L.; Hancock, 2012 and Dwivediet al, 2017. Biofilm matrices is another complexmechanism which is the phenotypeof three-dimensional acumulative gatheringof microbes since cells usuallycementstopolymeric substances in the networkof extracellular space, that contains majorly ofpolysaccharides, certainproteins, in addition to exteriornucleic acids.The biofilm is a permanentphenotypethat supports the microbe with capability to counterand drugsand antibiotics(Cho et al, 2018andPandeyet al, 2022). The formation of biofilm by multidrug-resistant microbes are correlatedto high tolerance of antibioticsbybacteria(Baptistaet al, 2018). The β-lactamase enzymeacts essentiallyfor damagingthe β-lactam ring in a β-lactam group of antibiotics. So, efflux pump, porins, biofilm and β-lactamase are themainphenotypes involving in developing MDRinmicrobes (Hancock, 2012 and Dwivediet al, 2017, Vergalliet al, 2020andPandeyet al, 2022). Once theantibiotic resistance suppressesthe antibiotic inhibitory effectonpathogenic bacteria, these resistant bacteria proliferatewith the treatment by antibiotics(Adegokeet al, 2016 and Breijyehet al, 2020).
MDR pathogens are serious and outstanding risks globally, which arises urgent needs for novel bioeffective substitutional for fighting aggressive MDR attacks(Al-Saggaf, 2022). Thus, the antibiotic resistance issueis one of the majorpublic healthproblems. (Baptistaet al, 2018).Since thehealth of humans is dramaticallyinfluencedby thecriticalriseofthe resistancemodalitiesof antimicrobials against detrimentalbacteria.Based on the declarationby World Health Organization (WHO) and CDC (Centers for Disease Control and Prevention)that the world is going toa post-antibiotic era, and predictionofmortality caused by contagious bacteria has risenin comparison to cancer(Baptistaet al, 2018and Munir, andAhmad, 2022). Additionally, according to WHO, the world load of healthcare-related infections varies amidst (7% - 12%). So, screen of Device Associated Nosocomial Infections (DANIs) in healthcare providers is an important issue. Shahbaz et al, (2022)used nanotechnology strategy to estimate the prevalence of microbial contaminations in DANIs patients and evaluated in vitroprevention of MDR bacterial strains. The increasein antibiotic resistance andthe deficiency of novel antimicrobialdrugattracts manyinitiatives to develop higher effective antimicrobial strategies for developingnew drugs,delivery systems, and targetingmanagement.
Severalstrategiesare developed to cope MDR. Nanoparticles(1–100 nm) have been applied asantimicrobialnovel drugto act as antimicrobial effective agentsor as delivery systems to bacterial infection sites. Nanoparticles such ascarbon nanotubes, inorganic and organic may abolishdrug resistance phenotypein bacteria. The nanotechnologyhas associated with itsantimicrobial activity and suppressing biofilm accumulative(Baptista et al, 2018). In last recent years, nanocomposite as a biocompatible oil-in-water cross-linked polymer has been developed, in which the nanocomposite degrades in some physiological environment demonstrating its ability to enter, eliminate wide-spectrum of MDR bacteria, expel biofilms without toxic side effect in vitro study of fibroblast cell line of mammalian. It was noticed that sequent passaging prevented bacteria from developing resistance to the used nanocomposites giving promising light for degradable nanocomposite effect on MDR microbes(Ryanet al, 2018).Another strategy isthe combined use of nanoparticles with plant-based antimicrobial to beatpossible toxic effect and to suppressthe resistance modalities by bacteria includingefflux pumps,formation of biofilms, mediation toquorum sensing; and for probablyplasmid treatment (Baptistaet al, 2018).Although the therapeuticnanotechnology against MDR introduces promising results but it is stillhaving challenges. So, many studies have been conducted to investigate nanomaterials as a newtherapeutic system to fight MDRmicrobes. Pandeyet al, (2022)synthesisednanomaterials, studied the wayof drug resistance in superbug P. aeruginosa, evaluatedthe nanocomposites as an anti-pseudomonal factor, and to act as a drug-resistant reversal tool, in addition tothe mode of action of these composites, and the nanomaterials’s druggability.Another antimicrobial nanotechnology was to usefungal culture of (Mucor circinelloides) and extract extract chitosanand convertit to nanochitosan to synthesize selenium bioactive nanoparticles (SeNPs)immediately utilizing extract of Hibiscus sabdariffa (Hb), and to evaluatetheir biocidal actions against MDR bacterial pathogens. The chitosanof fungi had 86.71% deacetylation and convertedactuallyto nanochitosan with size of 67.6 nm. The SeNPs have beenbiosynthesized directly bydexterityinteractionwith Hb, the average size of Hb/SeNPswas 12.1 nm. The couplingbetween nanochitosan and Hb/SeNPs was successful.The fabricated nanocomposites showedincreased antibacterialeffectagainstall studied MDR infectious microbes; nanochitosan /Hb/SeNPs composite wasveryrobustwith the largest growth inhibitingzones withminimumdosesof bactericidal. The structureof nanochitosan with Hb-biologically synthesized SeNPs resulted in potent concept of integrated bactericidal towards MDR microbes with biosafety, eco-cares, and efficienty. A very recent study has been designed to figure out the antibacterial effect of a nanosheet complex compound, and zinc oxide nanoparticles solely and also mixed with specific antibiotics against Pseudomonas aeruginosa isolate. These nanocompound suppressed the MIC of tetracycline from 16 to 64 times against MDR clinical isolate. The mode of action of the nanosheet was two synergistic effects includes firstly: interfered with efflux pumps, secondly: blocked biofilm synthesis. Furthermore, these nanosheet and nanocomposites decreased the mutant prevention concentration of TET (Pandey et al, 2022).
The emerging application of nanoparticles can be applied against mortal bacterial infections. So, different notion of nanomaterial technology can overcome the challenge of antibiotic resistance. Use of nanoparticles assist in the invention of fabricated antimicrobial nanotherapeutics by affirming of the functionality and delivery of nanoparticle’s exterior design and fabrication for antimicrobial load (Munir and Ahmad, 2022). One recent study on 324 patients diagnosed with DANI. Biosynthesized nanocomposite was analyzed for their antimicrobial activity Total 369 bacterial pathogens have been isolated from DANI patients. A ratio of 87% of these microbes was gram-negative bacilli and all were MDR. 41.5% of gram-negative isolates were ESBL maker. Among gram-positive bacteria, Methicillin-resistant Staphylococcus aureus represents a ratio of 7.3% of overall isolates. Nanocomposite exhibited dose-depending effect, since 100% bactericidal effect at a concentration of 5 mg/ml during 3 hours in incubation media, while less treatment with 2.5 mg/ml spends 6 hours to suppress perfect growth. Nanocomposites was an alternate treatment to block the DANIs on Acinetobacter baumannii and Citrobacter, the maximum causative species (Shahbaz et al, (2022).”
- C. auris, the most important multidrug resistant fungus in the study, which was proved practically by the presence of drug resistant mutations for its multidrug resistance. Also, in the present study, we have designed Ag-silicalite-1 nano-formulation to act as potential and effective nano-formulation to block the biofilm produced by the emerging drug-resistant auris.The authors isolated nineteen C. aurisstrains which were found in the hospitals. As the biofilm is one of the modes of action used by the microbe to resist the antibiotics. So, Ag-silicalite-1 was synthesized, characterized with different physicochemical techniques involving phase determination, surface texture, template effect and morphological characterizations to be applied as a potential anti-biofilm agent against C. auris.
With the increased prevalence of microbial contaminations in DANIs patients that caused by many microbes, in which C. auris, the most important multidrug resistant fungus. Then the anti-biofilm activities of designed Ag-silicalite-1 nano-formulation can be used as surface sterilizer against C. aurisin the future. As it is known that C. auriscontaminates many surfaces, devices and tools in centers of health care providers and increase its infectious effect among the already or new admitted patients and worsens their status. Only few number works studied on C. auris the most virulent infectious microbe. Therefore, this study isolated it from different hospitals and treated it with novel nano- formulation of Ag-silicalite-1 as antifungal agent against the isolated nineteenC. aurisstrains. The synthesized nanomaterials, 4wt%Ag/TiZSM-5 and Ag-silicalite-1 were tested for its antibiofilm activity. All the nineteen C. aurisstrains showed a significant inhibition on biofilm survival rate, with a lowest of 10% with Ag-silicalite-1 at 24 and 48h incubation. Profound morphogenesis and reduction in a number of C. auriscells were found by SEM and light microscopy. The presence of high surface area and uniform dispersion of nanosized Ag species displays an enhanced antifungal activity and therefore having a great potential against the emerging drug-resistant C. auris.It is clearly revealed that 4wt%Ag/TiZSM-5 and Ag-silicalite-1 has anti-fungal time and dose dependent effect.
The following references have also been added
Khardori, N.; Stevaux, C.; Ripley, K. Antibiotics: From the Beginning to the Future: Part 2. Indian J. Pediatr. 2020, 87, 43–47.
Mohr, K.I. History of Antibiotics Research. Curr. Top Microbiol. Immunol. 2016, 398, 237–272.
Mirzaei, B.; Bazgir, Z.N.; Goli, H.R.; Iranpour, F.; Mohammadi, F.; Babaei, R. Prevalence of Multi-Drug Resistant (MDR) and Extensively Drug-Resistant (XDR) Phenotypes of Pseudomonas Aeruginosa and Acinetobacter Baumannii Isolated in Clinical Samples from Northeast of Iran. BMC Res. Notes 2020, 13, 380.
Karaiskos, I.; Lagou, S.; Pontikis, K.; Rapti, V.; Poulakou, G. The “Old” and the “New” Antibiotics for MDR Gram-Negative Pathogens: For Whom, When, and How. Front. Public Health 2019, 7, 151.
Martins, M.; McCusker, M.P.; Viveiros, M.; Couto, I.; Fanning, S.; Pagès, J.-M.; Amaral, L. A Simple Method for Assessment of MDR Bacteria for Over-Expressed Efflux Pumps. Open Microbiol. J. 2013, 7, 72–82.
Baptista PV, McCusker MP, Carvalho A, et al. Nano-Strategies to Fight Multidrug Resistant Bacteria-"A Battle of the Titans". Front Microbiol. 2018;9:1441. Published 2018 Jul 2. doi:10.3389/fmicb.2018.01441
Xavier, D.E.; Picão, R.C.; Girardello, R.; Fehlberg, L.C.C.; Gales, A.C. Efflux Pumps Expression and Its Association with Porin Down-Regulation and Beta-Lactamase Production among Pseudomonas Aeruginosa Causing Bloodstream Infections in Brazil. BMC Microbiol. 2010, 10, 217.
Fernández, L.; Hancock, R.E.W. Adaptive and Mutational Resistance: Role of Porins and Efflux Pumps in Drug Resistance. Clin. Microbiol. Rev. 2012, 25, 661–681.
Dwivedi, G.R.; Singh, D.P.; Sharma, S.A.; Darokar, M.P. Efflux pumps: Warheads of gram-negative bacteria and efflux pump inhibitors. In New Approaches in Biological Research; Nova Science Publishers: New York, NY, USA, 2017; ISBN 978-1-5361-2115-5.
Cho, H.H.; Kwon, K.C.; Kim, S.; Park, Y.; Koo, S.H. Association between Biofilm Formation and Antimicrobial Resistance in Carbapenem-Resistant Pseudomonas Aeruginosa. Ann. Clin. Lab. Sci. 2018, 48, 363–368.
Vergalli, J.; Bodrenko, I.V.; Masi, M.; Moynié, L.; Acosta-Gutiérrez, S.; Naismith, J.H.; Davin-Regli, A.; Ceccarelli, M.; van den Berg, B.; Winterhalter, M.; et al. Porins and Small-Molecule Translocation across the Outer Membrane of Gram-Negative Bacteria. Nat. Rev. Microbiol. 2020, 18, 164–176.
Breijyeh, Z.; Jubeh, B.; Karaman, R. Resistance of Gram-Negative Bacteria to Current Antibacterial Agents and Approaches to Resolve It. Molecules 2020, 25, 1340.
Adegoke, A.A.; Faleye, A.C.; Singh, G.; Stenström, T.A. Antibiotic Resistant Superbugs: Assessment of the Interrelationship of Occurrence in Clinical Settings and Environmental Niches. Molecules 2016, 22, 29.
Shahbaz Aman, Divya Mittal, Hardeep Singh Tuli, Shubham Chauhan, Pardeep Singh, Sheetal Sharma, Reena V. Saini, Narinder Kaur, Adesh K. Saini,
Prevalence of multidrug-resistant strains in device associated nosocomial infection and their in vitro killing by nanocomposites, Annals of Medicine and Surgery,
2022, 103687, ISSN 2049-0801, https://doi.org/10.1016/j.amsu.2022.103687.
Mohammed S. Al-Saggaf, "Nanoconjugation between Fungal Nanochitosan and Biosynthesized Selenium Nanoparticles with Hibiscus sabdariffa Extract for Effectual Control of Multidrug-Resistant Bacteria", Journal of Nanomaterials, vol. 2022, Article ID 7583032, 9 pages, 2022. https://doi.org/10.1155/2022/7583032
Ryan F. Landis, Cheng-Hsuan Li, Akash Gupta, Yi-Wei Lee, Mahdieh Yazdani, Nipaporn Ngernyuang, Ismail Altinbasak, Sanaa Mansoor, Muhammadaha A. S. Khichi, Amitav Sanyal, and Vincent M. Rotello. Biodegradable Nanocomposite Antimicrobials for the Eradication of Multidrug-Resistant Bacterial Biofilms without Accumulated Resistance. Journal of the American Chemical Society 2018 140 (19), 6176-6182. DOI: 10.1021/jacs.8b03575
Pandey, P.; Sahoo, R.; Singh, K.; Pati, S.; Mathew, J.; Pandey, A.C.; Kant, R.; Han, I.; Choi, E.-H.; Dwivedi, G.R.; Yadav, D.K. Drug Resistance Reversal Potential of Nanoparticles/Nanocomposites via Antibiotic’s Potentiation in Multi Drug Resistant P. aeruginosa. Nanomaterials 2022, 12, 117. https://doi.org/10.3390/nano12010117
Mohammed S. Al-Saggaf, "Nanoconjugation between Fungal Nanochitosan and Biosynthesized Selenium Nanoparticles with Hibiscus sabdariffa Extract for Effectual Control of Multidrug-Resistant Bacteria", Journal of Nanomaterials, vol. 2022, Article ID 7583032, 9 pages, 2022. https://doi.org/10.1155/2022/7583032
Munir, M.U.; Ahmad, M.M. Nanomaterials Aiming to Tackle Antibiotic-Resistant Bacteria. Pharmaceutics 2022, 14, 582. https://doi.org/10.3390/pharmaceutics14030582
Shahbaz Aman, Divya Mittal, Hardeep Singh Tuli, Shubham Chauhan, Pardeep Singh, Sheetal Sharma, Reena V. Saini, Narinder Kaur, Adesh K. Saini, Prevalence of multidrug-resistant strains in device associated nosocomial infection and their in vitro killing by nanocomposites, Annals of Medicine and Surgery, 2022, 103687, ISSN 2049-0801, https://doi.org/10.1016/j.amsu.2022.103687.
- Moreover, there are too many low-level errors, such as different fonts (first paragraph of Intro), missing spaces (last paragraph of intro) etc. that should not appear in a submission.
Response:
Format errors have been removed in the revised manuscript
Materials & methods
- This section offers all the info needed for reproducing the results. However, there are many typing errors, such as: 24 h, 24h or 24 hrs??? Is it 37°C or 37 °C??? The font is not the same in every part.
Response:
Typographical errors have been removed in the revised manuscript and “°C” and “hours” were amended accordingly
Moving on to the Results & Discussion part,
- The “Collection of isolates” is suggested to be incorporated in the Materials and method part.
Response:
Collection of isolates has been removed from the Results & Discussion
- the figures 1,2,3,4,5 are not clearly shown. The data presented cannot be interpreted and the manuscript lacks a discussion about those results.
Response:
High definition pictures have been incorporated for high clarity.
The following details has been added
“The phylogenic analysis via branching diagram or a tree of 18S rRNA gene sequences with the standard sequences clearly indicates the evolutionary relationships among various Candiaspecies with the study isolate of C. aurisbased upon similarities in their 18S rRNA gene sequences.”
- The “Synthesis and characterization” part is obviously written from another person; thus, the results are presented in different manner in one paper. The authors used many techniques to characterize their samples, but they did not explain their results appropriately. For example, they used EDX mapping analysis and they did not highlighted in that section the uniformly dispersed Ag NPs.
Response:
Thank you for the great observation. Details have been added in the revised manuscript accordingly.
- On the other hand, in Figure 10 the arrows show Ag NPs, which are not uniformly dispersed. The authors need to add an explanation for that.
Response:
Morphological analysis using TEM shows that Ag nanoparticle dispersion is high over TiZSM-5 (Fig. 10c and d). This result correlate with the diffuse reflectance spectra of Ag/TiZSM-5 (Fig. 6C) due to reduction in peak attributed to Ag species with TiZSM-5 support (Fig. 6C). SEM-EDX mapping and TEM profile shows the presence of high dispersity (Fig. 9 and 10). In case of Ag-silicalite-1, the TEM image shows crystalline and some irregular aggregated particles on large crystals of silicalite-1 support. XRD pattern confirms that the AgNPs on silicalite-1 are of crystalline in nature.
- In the biofilm studies, the graphs presented in figure 11 need statistical analysis. Otherwise, expressions such as “depicted a significant inhibition” should be removed.
Response:
“depicted a significant inhibition” has been removed from the revised manuscript as per the suggestion by the reviewer.
- Why are two time points presented; Since the activity is similar between 24 h and 48 h, there is no reason for presenting both of them (move one of them in SI).
Response:
As per the suggestion by the reviewer the 48 hours has been moved to Supplementary Figure [1] and changes have been done in the legends as follows
Figure [11] Graph showing the effect of synthesized nanomaterial on the formation of biofilm of nineteen C. aurisstrains after 24 hours incubation period.
Supplementary Figure [1] Graph showing the effect of synthesized nanomaterial on the formation of biofilm of nineteen C. auris strains after 48 hours of incubation period
- The morphogenesis study by SEM needs arrows or something in the figures to highlight all the changes written in the text.
Response:
As per the suggestion by the reviewer changes have been done in the legends of the figure 10.
In the conclusion part,
- the significance of the study should be clearly written. This part presented in this manuscript looks like an abstract, not a conclusion.
Response:
The following conclusion has been added in the revised MS.
It is widely known that multidrug-resistant microbial contaminations in patients with device-associated nosocomial infections are prevalent and caused by a variety of microorganisms, including C. auris, the most common multidrug-resistant fungus. In the future, the anti-biofilm properties of the developed Ag-silicalite-1 nano-formulation might be exploited as a surface sterilizer against C. auris. We briefly discussed current advances in employing the aforesaid nanomaterials to attack C. aurisin antifungal applications in this paper. However, there is still more work to be done before these materials may be used in therapeutic settings. Nanoparticles' precise mechanism, biological effects, and toxic and side consequences remain unknown. We still don't know if nanoparticles and their metabolites are actually harmless to the environment and biology. To answer all of these issues, further laboratory and clinical trials are needed, but nanomaterials' promise as antifungal therapies is undeniable.
References
- Authors are suggested to include more updated references reflecting the diversity of antimicrobial nanomaterials and strategies against MDR microbes. Hence, readers can learn the research background comprehensively and the latest progress in the related field.
Response:
The following references have also been added to add more details about antimicrobial nanomaterials and strategies against MDR microbes
Wang LS, Gupta A & Rotello VM Nanomaterials for the treatment of bacterial biofilms. ACS Infect. Dis 2, 3–4 (2016).
Gupta A, Landis RF & Rotello VM Nanoparticle-based antimicrobials: Surface functionality is critical. F1000Res. 5, 364 (2016).
Vert M. Terminology for biorelated polymers and applications (IUPAC Recommendations 2012). Pure Appl. Chem 84, 377–410 (2012).
Makabenta JMV, Nabawy A, Li CH, Schmidt-Malan S, Patel R, Rotello VM. Nanomaterial-based therapeutics for antibiotic-resistant bacterial infections. Nat Rev Microbiol. 2021 Jan;19(1):23-36.
Gupta A, Mumtaz S, Li C-H, Hussain I & Rotello VM Combatting antibiotic-resistant bacteria using nanomaterials. Chem. Soc. Rev 48, 415–427 (2019)
Khardori, N.; Stevaux, C.; Ripley, K. Antibiotics: From the Beginning to the Future: Part 2. Indian J. Pediatr. 2020, 87, 43–47.
Mohr, K.I. History of Antibiotics Research. Curr. Top Microbiol. Immunol. 2016, 398, 237–272.
Mirzaei, B.; Bazgir, Z.N.; Goli, H.R.; Iranpour, F.; Mohammadi, F.; Babaei, R. Prevalence of Multi-Drug Resistant (MDR) and Extensively Drug-Resistant (XDR) Phenotypes of Pseudomonas Aeruginosa and Acinetobacter Baumannii Isolated in Clinical Samples from Northeast of Iran. BMC Res. Notes 2020, 13, 380.
Karaiskos, I.; Lagou, S.; Pontikis, K.; Rapti, V.; Poulakou, G. The “Old” and the “New” Antibiotics for MDR Gram-Negative Pathogens: For Whom, When, and How. Front. Public Health 2019, 7, 151.
Martins, M.; McCusker, M.P.; Viveiros, M.; Couto, I.; Fanning, S.; Pagès, J.-M.; Amaral, L. A Simple Method for Assessment of MDR Bacteria for Over-Expressed Efflux Pumps. Open Microbiol. J. 2013, 7, 72–82.
Baptista PV, McCusker MP, Carvalho A, et al. Nano-Strategies to Fight Multidrug Resistant Bacteria-"A Battle of the Titans". Front Microbiol. 2018;9:1441. Published 2018 Jul 2. doi:10.3389/fmicb.2018.01441
Xavier, D.E.; Picão, R.C.; Girardello, R.; Fehlberg, L.C.C.; Gales, A.C. Efflux Pumps Expression and Its Association with Porin Down-Regulation and Beta-Lactamase Production among Pseudomonas Aeruginosa Causing Bloodstream Infections in Brazil. BMC Microbiol. 2010, 10, 217.
Fernández, L.; Hancock, R.E.W. Adaptive and Mutational Resistance: Role of Porins and Efflux Pumps in Drug Resistance. Clin. Microbiol. Rev. 2012, 25, 661–681.
Dwivedi, G.R.; Singh, D.P.; Sharma, S.A.; Darokar, M.P. Efflux pumps: Warheads of gram-negative bacteria and efflux pump inhibitors. In New Approaches in Biological Research; Nova Science Publishers: New York, NY, USA, 2017; ISBN 978-1-5361-2115-5.
Cho, H.H.; Kwon, K.C.; Kim, S.; Park, Y.; Koo, S.H. Association between Biofilm Formation and Antimicrobial Resistance in Carbapenem-Resistant Pseudomonas Aeruginosa. Ann. Clin. Lab. Sci. 2018, 48, 363–368.
Vergalli, J.; Bodrenko, I.V.; Masi, M.; Moynié, L.; Acosta-Gutiérrez, S.; Naismith, J.H.; Davin-Regli, A.; Ceccarelli, M.; van den Berg, B.; Winterhalter, M.; et al. Porins and Small-Molecule Translocation across the Outer Membrane of Gram-Negative Bacteria. Nat. Rev. Microbiol. 2020, 18, 164–176.
Breijyeh, Z.; Jubeh, B.; Karaman, R. Resistance of Gram-Negative Bacteria to Current Antibacterial Agents and Approaches to Resolve It. Molecules 2020, 25, 1340.
Adegoke, A.A.; Faleye, A.C.; Singh, G.; Stenström, T.A. Antibiotic Resistant Superbugs: Assessment of the Interrelationship of Occurrence in Clinical Settings and Environmental Niches. Molecules 2016, 22, 29.
Shahbaz Aman, Divya Mittal, Hardeep Singh Tuli, Shubham Chauhan, Pardeep Singh, Sheetal Sharma, Reena V. Saini, Narinder Kaur, Adesh K. Saini,
Prevalence of multidrug-resistant strains in device associated nosocomial infection and their in vitro killing by nanocomposites, Annals of Medicine and Surgery,
2022, 103687, ISSN 2049-0801, https://doi.org/10.1016/j.amsu.2022.103687.
Mohammed S. Al-Saggaf, "Nanoconjugation between Fungal Nanochitosan and Biosynthesized Selenium Nanoparticles with Hibiscus sabdariffa Extract for Effectual Control of Multidrug-Resistant Bacteria", Journal of Nanomaterials, vol. 2022, Article ID 7583032, 9 pages, 2022. https://doi.org/10.1155/2022/7583032
Ryan F. Landis, Cheng-Hsuan Li, Akash Gupta, Yi-Wei Lee, Mahdieh Yazdani, Nipaporn Ngernyuang, Ismail Altinbasak, Sanaa Mansoor, Muhammadaha A. S. Khichi, Amitav Sanyal, and Vincent M. Rotello. Biodegradable Nanocomposite Antimicrobials for the Eradication of Multidrug-Resistant Bacterial Biofilms without Accumulated Resistance. Journal of the American Chemical Society 2018 140 (19), 6176-6182. DOI: 10.1021/jacs.8b03575
Pandey, P.; Sahoo, R.; Singh, K.; Pati, S.; Mathew, J.; Pandey, A.C.; Kant, R.; Han, I.; Choi, E.-H.; Dwivedi, G.R.; Yadav, D.K. Drug Resistance Reversal Potential of Nanoparticles/Nanocomposites via Antibiotic’s Potentiation in Multi Drug Resistant P. aeruginosa. Nanomaterials 2022, 12, 117. https://doi.org/10.3390/nano12010117
Mohammed S. Al-Saggaf, "Nanoconjugation between Fungal Nanochitosan and Biosynthesized Selenium Nanoparticles with Hibiscus sabdariffa Extract for Effectual Control of Multidrug-Resistant Bacteria", Journal of Nanomaterials, vol. 2022, Article ID 7583032, 9 pages, 2022. https://doi.org/10.1155/2022/7583032
Munir, M.U.; Ahmad, M.M. Nanomaterials Aiming to Tackle Antibiotic-Resistant Bacteria. Pharmaceutics 2022, 14, 582. https://doi.org/10.3390/pharmaceutics14030582
Shahbaz Aman, Divya Mittal, Hardeep Singh Tuli, Shubham Chauhan, Pardeep Singh, Sheetal Sharma, Reena V. Saini, Narinder Kaur, Adesh K. Saini, Prevalence of multidrug-resistant strains in device associated nosocomial infection and their in vitro killing by nanocomposites, Annals of Medicine and Surgery, 2022, 103687, ISSN 2049-0801, https://doi.org/10.1016/j.amsu.2022.103687.

Reviewer 2 Report
General Remarks: The topic is interesting. The title is in accordance with the subject dealt with in the manuscript. The abstract expresses the objectives and main results obtained in the study. In my opinion the authors have tried to put their findings into context and many of my suggestions are for clarifications. The major points they should consider are the following:
- Reference numbers cited in the text are not separated by commas, please check it and change it to the correct form
- The letters in the text have different size and format, please check it, and change it to the correct form.
- In my opinion, including in the Abstract all the characterization techniques used is not appropriate. I consider that the paragraph can be re-written only put: The prophylactic and therapeutic application of hydrothermally synthesized Ag-silicalite-1 (Si/Ag ratio 25) nanomaterial against the clinical strains of C. auris were tested. TiZSM-5, 4wt%Ag/TiZSM-5 were prepared using impregnation technique, Ag-silicalite-1 (25) and Ag-silicalite-1 (100) nano formulations were characterized using different techniques. And explain these techniques with more details in the text of the article (Introduction, Materials and Methods or Results and Discussion section.
- Results and Discussion.
-Collection of isolates section must be in the section of Materials and methods and not into Results and Discussion section.
-As I understand it, according to previous tests carried out, 18S RNA is used as the housekeeping standard for PCR samples, however, the way in which the results are explained both in the text of the article and in the abstract seems to indicate that it has been used as a specific marker for sample characterization. If so, could the authors explain why? Otherwise, I consider it necessary to change the way it is written in the text so as not to confuse readers.
-In my opinion, the images should be in a 2D format and saved in a higher quality format that makes it easier for the reader to see the details.
Discussion. There was very little discussion of how this study fits into the context of other similar studies, or in what way it contributed to the field beyond what other studies have shown.
Other comments:
- Results and Discussion section is limited to a mere presentation of the results, without comparing them with the published bibliography on the subject or explaining how these results go beyond the state of the art.
-Conclusions section is more like a discussion than a real conclusion of the work, in terms of summarizing the results obtained and demonstrating their scope.
Author Response
Reviewer 2
General Remarks: The topic is interesting. The title is in accordance with the subject dealt with in the manuscript. The abstract expresses the objectives and main results obtained in the study. In my opinion the authors have tried to put their findings into context and many of my suggestions are for clarifications. The major points they should consider are the following:
Response:
Authors thank the second reviewer for the valuable suggestions and comments. These suggestion are very helpful to improve our manuscript. Responses for each comments are presented below.
- Reference numbers cited in the text are not separated by commas, please check it and change it to the correct form
Response:
As per the suggestion by the reviewer, authors have ensured the reference numbers cited in the text are separated by commas.
- The letters in the text have different size and format, please check it, and change it to the correct form.
Response:
As per the suggestion by the reviewer, authors have ensured the text are in same size and format.
- In my opinion, including in the Abstract all the characterization techniques used is not appropriate. I consider that the paragraph can be re-written only put: The prophylactic and therapeutic application of hydrothermally synthesized Ag-silicalite-1 (Si/Ag ratio 25) nanomaterial against the clinical strains of C. auris were tested. TiZSM-5, 4wt%Ag/TiZSM-5 were prepared using impregnation technique, Ag-silicalite-1 (25) and Ag-silicalite-1 (100) nano formulations were characterized using different techniques. And explain these techniques with more details in the text of the article (Introduction, Materials and Methods or Results and Discussion section.
Response:
As per the suggestion by the reviewer, authors have revised the following statement in the abstract.
“The prophylactic and therapeutic application of hydrothermally synthesized Ag-silicalite-1 (Si/Ag ratio 25) nanomaterial against the clinical strains of C. auriswere tested.TiZSM-5, 4wt%Ag/TiZSM-5 were prepared using impregnation technique, Ag-silicalite-1 (25) and Ag-silicalite-1 (100) nano formulations were characterized using different phases determination (XRD), surface area analysis (BET), diffuse reflectance UV-Visible spectroscopy (DRS-UV-Vis), thermogravimetric analysis and differential thermal analysis (TGA-DTA), morphological studies [scanning electron microscopy/energy dispersive X-Rays spectroscopy (SEM/EDX) and transmission electron microscope (TEM)]”
Has been replaced as follwos
“The prophylactic and therapeutic application of hydrothermally synthesized Ag-silicalite-1 (Si/Ag ratio 25) nanomaterial against the clinical strains of C. auriswere tested. TiZSM-5, 4wt%Ag/TiZSM-5 were prepared using impregnation technique, Ag-silicalite-1 (25) and Ag-silicalite-1 (100) nano formulations were characterized using different techniques”
- Results and Discussion.
-Collection of isolates section must be in the section of Materials and methods and not into Results and Discussion section.
Response:
Collection of isolates has been removed from the Results & Discussion
-As I understand it, according to previous tests carried out, 18S RNA is used as the housekeeping standard for PCR samples, however, the way in which the results are explained both in the text of the article and in the abstract seems to indicate that it has been used as a specific marker for sample characterization. If so, could the authors explain why? Otherwise, I consider it necessary to change the way it is written in the text so as not to confuse readers.
Response:
As per the suggestion by the reviewer, the following details have been added in the revised manuscript.
“The phylogenic analysis via branching diagram or a tree of 18S rRNA gene sequences with the standard sequences clearly indicates the evolutionary relationships among various Candiaspecies with the study isolate of C. aurisbased upon similarities in their 18S rRNAgene sequences.”
-In my opinion, the images should be in a 2D format and saved in a higher quality format that makes it easier for the reader to see the details.
Response:
3D effect of the pictures have been removed and the 2D format for the higher quality pictures were ensured in the revised manuscript.
Discussion. There was very little discussion of how this study fits into the context of other similar studies, or in what way it contributed to the field beyond what other studies have shown.
Response:
As per the suggestion by the reviewer, the following details on other similar studies were added in the revised manuscript. Related references were also added
“The mechanism of action of Ag-silicalite-1 nanoformulation on drug-resistant C. aurisas an effective antibiofilm agent may be due to unique properties of nanomaterials. The unique properties of nanomaterials which include size, shape, and surface chemistry generally produce significant inhibitory action on the bacteria. It has been reported that different sizes and shapes of nanomaterials are analogous to bacterial bio-molecular components, and enhances the better connections that can be controlled through surface functionalization. Moreover, high surface to volume ratios of the nanomaterials and multivalent interactions are important aspects for creating antibacterial agents 43, 44, 45. Nanomaterials use multiple bactericidal routes and mechanisms which include direct bacterial cell wall damage, and also generation of reactive oxygen species (ROS) and binding to intra-cellular bacterial constituents to be successfully kill the bacteria 46. The mechanisms by which nanomaterials interact with bacteria may due to unique physicochemical properties, especially multi-valent interactions with bacterial cells. It has been suggested that Van der Waals forces, receptor-ligand, hydrophobic interactions and electrostatic attractions play a role in nanomaterial-bacteria interfaces for the robust antibacterial action 47.”
- Wang LS, Gupta A & Rotello VM Nanomaterials for the treatment of bacterial biofilms. ACS Infect. Dis 2, 3–4 (2016).
- Gupta A, Landis RF & Rotello VM Nanoparticle-based antimicrobials: Surface functionality is critical. F1000Res. 5, 364 (2016).
- Vert M. Terminology for biorelated polymers and applications (IUPAC Recommendations 2012). Pure Appl. Chem 84, 377–410 (2012).
- Makabenta JMV, Nabawy A, Li CH, Schmidt-Malan S, Patel R, Rotello VM. Nanomaterial-based therapeutics for antibiotic-resistant bacterial infections. Nat Rev Microbiol. 2021 Jan;19(1):23-36.
- Gupta A, Mumtaz S, Li C-H, Hussain I & Rotello VM Combatting antibiotic-resistant bacteria using nanomaterials. Chem. Soc. Rev 48, 415–427 (2019)
Other comments:
- Results and Discussion section is limited to a mere presentation of the results, without comparing them with the published bibliography on the subject or explaining how these results go beyond the state of the art.
Response:
As per the suggestion by the reviewer, more details were added in the revised manuscript. Related references were also added.
-Conclusions section is more like a discussion than a real conclusion of the work, in terms of summarizing the results obtained and demonstrating their scope.
Response:
The following conclusion has been added in the revised MS.
It is widely known that multidrug-resistant microbial contaminations in patients with device-associated nosocomial infections are prevalent and caused by a variety of microorganisms, including C. auris, the most common multidrug-resistant fungus. In the future, the anti-biofilm properties of the developed Ag-silicalite-1 nano-formulation might be exploited as a surface sterilizer against C. auris. We briefly discussed current advances in employing the aforesaid nanomaterials to attack C. aurisin antifungal applications in this paper. However, there is still more work to be done before these materials may be used in therapeutic settings. Nanoparticles' precise mechanism, biological effects, and toxic and side consequences remain unknown. We still don't know if nanoparticles and their metabolites are actually harmless to the environment and biology. To answer all of these issues, further laboratory and clinical trials are needed, but nanomaterials' promise as antifungal therapies is undeniable.
Reviewer 3 Report
In this study, 19 Candida auris strains were collected and identified by sequencing. Mutations associated with drug resistance in the ERG11, TAC1B, and FUR1 genes were screened. Then Ag-silicalite-1 (Si/Ag ratio of 25) particles were prepared by hydrothermal synthesis, characterized, and the inhibitory effect on C. auris strains was evaluated. In general, this work seems to be useful, but some major concerns still need to be addressed before it is suitable for publication in this journal.
- The authors screened out the gene mutation of the strain, but what is the relationship between this mutation study and the subsequent antibacterial research of nanoparticles? The two parts seem to be separated.
- In the antibiofilm study, Ag-Siliclate-1 and 4wt% Ag/TiZSM-5 have different antibacterial effects in different strains. What is the reason for this difference? Is it related to the structure and composition of nanomaterials?
- The author should carefully check the Figure legend in the manuscript. Such as in Page 16 “The results of TEM are displayed by Fig. 5 under different magnifications”. The Figure number was confusing.
- The author should add SD value in Fig 8(b). (d). (f) and Fig11.
- The author should explain the difference between Fig 10 a and Fig 10b, fig10 g-h in the legend.

Author Response
Reviewer 3
In this study, 19 Candida auris strains were collected and identified by sequencing. Mutations associated with drug resistance in the ERG11, TAC1B, and FUR1 genes were screened. Then Ag-silicalite-1 (Si/Ag ratio of 25) particles were prepared by hydrothermal synthesis, characterized, and the inhibitory effect on C. auris strains was evaluated. In general, this work seems to be useful, but some major concerns still need to be addressed before it is suitable for publication in this journal.
Response:
The authors thank the third reviewer for taking the time to read our paper and provide helpful feedback. These suggestions are quite beneficial in improving our manuscript. The responses to each of the remarks are included below.
- The authors screened out the gene mutation of the strain, but what is the relationship between this mutation study and the subsequent antibacterial research of nanoparticles? The two parts seem to be separated.
Response:
The study substantially proved the presence of drug resistant mutations to ensure the multi drug resistant nature of the collected strains. Which was not proved in most the nanomaterial based studies. Reason for the differences on the formation of biofilm of nineteen C. aurisdue to nanomaterial still need to be studied more in depth molecular level through whole genome and transcriptomic studies.
- In the antibiofilm study, Ag-Siliclate-1 and 4wt% Ag/TiZSM-5 have different antibacterial effects in different strains. What is the reason for this difference? Is it related to the structure and composition of nanomaterials?
Response:
Fig. 6A. XRD of (a) TiZSM-5, (b) 4wtAg/TiZSM-5 and (c) Ag-silicalite-1.
Thank you, reviewer, for this query. As stated, we have observed a predominant inhibition of 89% with Ag-silicalite-1 compared to 4wt%Ag/TiZSM-5 with inhibition around 30%-50%. It is clearly shown that TiZSM-5 had no inhibitory effect on neither the biofilm nor the planktonic cells except one isolate. Therefore, the activity difference is primarily attributed to the dispersion state and particle size of AgNPs on silicalite-1 and TiZSM-5. To detect the nature of AgNPs species on two supports, we analyzed the XRD pattern of three samples between 2 theta ranging 30-60 °(Fig. 6Aa-c). X-ray diffraction analysis shows that Ag nanoparticle dispersion occurs on TiZSM-5 and is less than 10 nm (below detection limit of XRD) over TiZSM-5 (Fig 6Aa and b). There are no chunks like Ag particles was observed on the external surface of TiZSM-5 indicating high dispersion state of Ag on TiZSM-5. This result correlate with the diffuse reflectance spectra of Ag/TiZSM-5 (Fig. 6C) due to reduction in peak attributed to Ag species with TiZSM-5 support (Fig. 6C). SEM-EDX mapping and TEM profile shows the presence of high dispersity (Fig. 9 and 10). In case of Ag-silicalite-1, the XRD pattern confirms that the AgNPs are of crystalline in nature (Fig. 6Ac). Therefore, such crystalline AgNPs on such large crystal of Ag-silicalite-1 is shown to favor the structural and metabolic disruption in C. auriscells. Furthermore, some unknown reasons also we can’t neglect.
- The author should carefully check the Figure legend in the manuscript. Such as in Page 16 “The results of TEM are displayed by Fig. 5 under different magnifications”. The Figure number was confusing.
Response:
As per the suggestion by the reviewer changes have been done in the legends of the figures.
- The author should add SD value in Fig 8(b). (d). (f) and Fig11.
Response:
As per the suggestion by the other reviewers the 48 hours has been moved to Supplementary Figure [1] and changes have been done in the legends .
- The author should explain the difference between Fig 10 a and Fig 10b, fig10 g-h in the legend.
Response:
The legend in Fig. 10 has been revised to make sure details are explained in our revised manuscript as follows .
Fig 10. TEM images of support (a, b) TiZSM-5, Ag impregnated TiZSM-5 (c, d) 4wt%Ag/TiZSM-5 and (e-h) Ag-silicalite-1 at different scale bar of 500 nm and 100 nm.

Reviewer 4 Report
This manuscript investigated therapeutic intervention for various hospital setting strains of biofilm forming candida auris using nanomaterial Ag-silicalite-1 zeolite. Although the results might be reliable, it is lack of scientific soundness. For examples, Ag-silicalite-1 zeolite was not well characterized in terms of the dispersity and the amount of Ag, which is important to understand the origin of inhibition performance. Nineteen strains were collected, but it seems to make no sense to understand the interaction between Ag and the fungi at gene level. The resolution of Figs. 1 to 5 was poor, making them present nothing to readers. Fig. 9, SEM photos should be given to help to understand EDX mapping, and magnification was not enough to discuss the dispersity. Fig. 11, CA1-CA19, some data seem to be deficient, please supplement data the delete the strains without comprehensive study. In general, the manuscript was poorly organized and presented, and not suitable for publication at current stage.
Author Response
Reviewer 4
This manuscript investigated therapeutic intervention for various hospital setting strains of biofilm forming candida auris using nanomaterial Ag-silicalite-1 zeolite. Although the results might be reliable, it is lack of scientific soundness.
Response:
The authors appreciate the fourth reviewer's time and effort in reading our manuscript and providing constructive input. These suggestion are helpful to improve our article significantly. Below are the replies to each of the comments.
For examples, Ag-silicalite-1 zeolite was not well characterized in terms of the dispersity and the amount of Ag, which is important to understand the origin of inhibition performance.
Response:
Thank you. The additional discussion with respect to dispersity has been discussed in our revised manuscript. The detail of silver content determined was included in our revised manuscript.
Nineteen strains were collected, but it seems to make no sense to understand the interaction between Ag and the fungi at gene level. The resolution of Figs. 1 to 5 was poor, making them present nothing to readers.
Response:
The study substantially proved the presence of drug resistant mutations to ensure the multi drug resistant nature of the collected strains. Which was not proved in most the nanomaterial based studies. Reason for the differences on the formation of biofilm of nineteen C. aurisdue to nanomaterial still need to be studied more in depth molecular level. High resolution pictures have been added.
Fig. 9, SEM photos should be given to help to understand EDX mapping, and magnification was not enough to discuss the dispersity.
Response:
As per the suggestion by the reviewer changes have been done in the revised manuscript.
Fig. 11, CA1-CA19, some data seem to be deficient, please supplement data the delete the strains without comprehensive study.
Response:
As per the suggestion by the reviewer the 48 hours has been moved to Supplementary Figure [1] and changes have been done in the legends as follows
Figure [11] Graph showing the effect of synthesized nanomaterial on the formation of biofilm of nineteen C. aurisstrains after 24 hours incubation period.
Supplementary Figure [1] Graph showing the effect of synthesized nanomaterial on the formation of biofilm of nineteen C. auris strains after 48 hours of incubation period
In general, the manuscript was poorly organized and presented, and not suitable for publication at current stage.
Response:
Authors have sufficiently added more details and have revised the organisation of the manuscript. Hope the revised is considerable for publication.

Round 2
Reviewer 3 Report
This manuscript, despite some revisions by the authors, still has many problems. I don't think this article can be published in this version.
For example,
1. Although the author modified some serial numbers of the picture legend. However, there are two table1 in this manuscript. More importantly, the number of pictures in manuscripts is confusing, making it difficult for readers to understand. For example, The EDX spectrum of Ag-Silicalite was composed of Ag (4.65%), O (49.87%), Na (1.80%), and Si (33.13%) (Fig. 3f). What is Figure 3? There are many similar mistakes.
2. In the last version, I suggested adding the standard deviation in the figures or tables, but the authors don't seem to have made any changes. How many times were each experiment repeated? If it is 3 times, the authors should add the SD value and perform a statistical analysis.
3. In picture 10, I suggest explaining the difference between a,b a,c and e,f. But the author explains them as "difference of scale bar of 500 nm and 100 nm." Also, I don't see the picture do any updates?

Reviewer 4 Report
This manuscript was not well organized. The infomation was fragmented without clear logic and novelty. For examples, what can we learn from ninetween strains? what can we know from the correpondence of various strains and inhibitory effect? how to evaluate the dependence of inhibitory effect on the porous structure and the dispersity of Ag? Although many pictures were provided, most of them were not indicated and intepreted. For examples, Fig. 7, what did the yellow rectangles mean? how were Ag-silicates crystals identified? Fig. 10, I can not agree that Ag was well dispersed from TEM graphs. Fig. 11, why was it only two bars for CA14, CA18 and CA19? and Fig. 11 was not discussed at all. Fig. 14 did not provide more information than Fig. 11, to some extent. In general, I can not grasp the highlights and novelty of this manuscript. It is not suitable for publication at current form.